# Simple is Effective: The Roles of Graphs and Large Language Models in Knowledge-Graph-Based Retrieval-Augmented Generation

**Mufei Li**[\*], **Siqi Miao**[\*], **Pan Li**
 Georgia Institute of Technology
{mufei.li, siqi.miao, panli}@gatech.edu
[\*]Equal contribution.

## Abstract

Large Language Models (LLMs) demonstrate strong reasoning abilities but face limitations such as hallucinations and outdated knowledge. Knowledge Graph (KG)-based Retrieval-Augmented Generation (RAG) addresses these issues by grounding LLM outputs in structured external knowledge from KGs. However, current KG-based RAG frameworks still struggle to optimize the trade-off between retrieval effectiveness and efficiency in identifying a suitable amount of relevant graph information for the LLM to digest. We introduce SubgraphRAG, extending the KG-based RAG framework that retrieves subgraphs and leverages LLMs for reasoning and answer prediction. Our approach innovatively integrates a lightweight multilayer perceptron (MLP) with a parallel triple-scoring mechanism for efficient and flexible subgraph retrieval while encoding directional structural distances to enhance retrieval effectiveness. The size of retrieved subgraphs can be flexibly adjusted to match the query's needs and the downstream LLM's capabilities. This design strikes a balance between model complexity and reasoning power, enabling scalable and generalizable retrieval processes. Notably, based on our retrieved subgraphs, smaller LLMs like Llama3.1-8B-Instruct deliver competitive results with explainable reasoning, while larger models like GPT-4o achieve state-of-the-art accuracy compared with previous baselines—all without fine-tuning. Extensive evaluations on the WebQSP and CWQ benchmarks highlight SubgraphRAG's strengths in efficiency, accuracy, and reliability by reducing hallucinations and improving response grounding. Our implementation is available at https://github.com/Graph-COM/SubgraphRAG.

## 1 Introduction

Large language models (LLMs) have increasingly demonstrated remarkable capabilities (Brown et al., 2020; Kojima et al., 2022; Wei et al., 2022; Bubeck et al., 2023; Yao et al., 2023; Huang & Chang, 2023). However, issues like hallucinations (Ji et al., 2023; Huang et al., 2023; Zhang et al., 2023), outdated knowledge (Dhingra et al., 2022; Kasai et al., 2023), and a lack of domain expertise (Li et al., 2023b) undermine their trustworthiness. Retrieval-augmented generation (RAG) has emerged as a promising strategy to mitigate these problems by grounding LLM outputs in external knowledge sources (Shuster et al., 2021; Borgeaud et al., 2022; Vu et al., 2024; Gao et al., 2024b).

Despite the effectiveness of text-based RAG, graph-based structures offer a more efficient alternative for organizing knowledge (Chein & Mugnier, 2008). Graphs facilitate explicit representation of relationships, reduce information redundancy, and allow for more flexible updates (Robinson et al., 2015). Recent studies have explored using graph-structured knowledge, particularly knowledge graphs (KGs), as RAG resources (Pan et al., 2024; Peng et al., 2024; Edge et al., 2024). However, developing effective and efficient frameworks for KG-based RAG remains limited due to the unique challenges posed by information retrieval from KGs.

Firstly, traditional text retrieval methods, such as BM25 (Robertson et al., 1994; Robertson & Zaragoza, 2009) and dense retrieval (Karpukhin et al., 2020), are insufficient for supporting LLMs

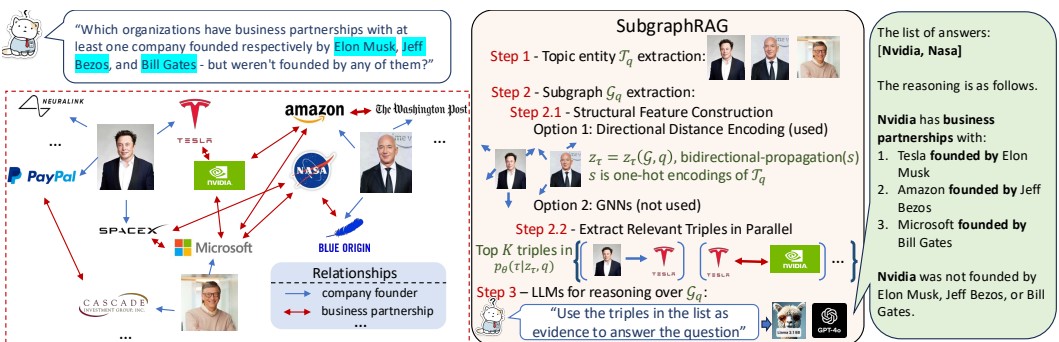

Figure 1: The SubgraphRAG framework. Retrieved subgraphs are flexible in form and size, consisting of triples extracted in parallel. This example subgraph has a complex form beyond a path.

in complex reasoning tasks (Sun et al., 2018). For instance, a query like "What is the most famous painting by a contemporary of Michelangelo and Raphael?" requires not only retrieving works by their contemporaries but also reasoning about relationships beyond Michelangelo and Raphael themselves. Thus, KG retrieval goes beyond basic entity linking, requiring the extraction of non-local entities connected through multi-hop and semantically relevant relationships to support reasoning (Jiang et al., 2023a; Luo et al., 2024b; Sun et al., 2024a). Such information is often best represented as KG subgraphs, whose retrieval enables more effective downstream reasoning.

Second, KG-based RAG faces significant computational challenges. In order to handle online queries and adapt to dynamic KG updates (Trivedi et al., 2017; Liang et al., 2024), efficiently identifying relevant structural information while meeting latency requirements is crucial for designing a practical KG-based RAG framework. Efficient approximate methods for nearest neighbor search like locality-sensitive hashing (Indyk & Motwani, 1998) are not suitable for structure extraction.

Third, retrieved structural information must cover the critical evidence needed to answer the query while respecting the reasoning capacity of LLMs. Simply expanding the context window increases computational cost and can degrade RAG performance due to the introduction of irrelevant information (Xu et al., 2024), causing the "lost in the middle" phenomenon (Liu et al., 2024b). To prevent these issues, redundant structural information should be pruned to keep only relevant evidence within the LLMs' processing limits, improving accuracy and avoiding hallucinations of LLMs.

Existing KG-based RAG frameworks often struggle to effectively balance retrieval with reasoning over the retrieved data, limiting their ability to address these challenges. For example, many approaches rely on multiple LLM calls for iterative KG search, introducing significant complexity for each query (Kim et al.; Gao et al., 2024a; Wang et al., 2024; Guo et al., 2024; Ma et al., 2024; Sun et al., 2024a; Jiang et al., 2024; Jin et al., 2024; Xiong et al., 2024). Furthermore, the mismatch between large KG search spaces and constrained LLM context windows poses challenges to the coverage of LLM-based retrieval. Conversely, some methods employ lighter models, such as Long Short-Term Memory or Graph Neural Networks, for both retrieval and reasoning (Zhang et al., 2022; Liu et al., 2024a; Sun et al., 2019). Despite being more efficient, these methods are constrained by the inferior reasoning capabilities of the lighter models. Additionally, certain approaches retrieve fixed types of subgraphs (e.g., paths) for convenience (Zhang et al., 2022; Luo et al., 2024b), which restricts the coverage of critical evidence for LLM reasoning—a point we explore further in Sec. 3.1.

**Design Principles** We argue that there is an inherent tradeoff between model complexity and reasoning capability. To effectively search over KGs, which are expected to grow rapidly, knowledge retrievers should remain lightweight, flexible, generalizable, and equipped with basic reasoning abilities to efficiently filter (roughly) relevant information from vast amounts of irrelevant data while delegating complex reasoning tasks to LLMs. As LLMs continue to demonstrate increasingly sophisticated reasoning capabilities and are likely to improve further, this division of labor will become more sensible. As long as the retrieved information fits within the LLM's reasoning capacity, LLMs—leveraging their superior reasoning power—can then perform more fine-grained analysis and provide accurate answers with appropriate prompting. This approach extends the two-stage Recall & Ranking (rough to fine) framework commonly used in traditional information retrieval and recommendation pipelines, with the key advance being that each stage is paired with appropriately tiered reasoning capabilities from AI models to meet the demands of answering complex queries.

Besides, for questions proven too challenging, the concept of iterating the above process can be adopted. However, this consideration is beyond the scope of the current work.

**Present Work** Our KG-based RAG framework, SubgraphRAG (Fig. 1), follows a pipeline that first retrieves a relevant subgraph and then employs LLMs to reason over it. While this approach mirrors some existing methods (He et al., 2021; Jiang et al., 2023b), SubgraphRAG introduces novel design elements that significantly improve both efficiency and effectiveness by adhering to the aforementioned principles. For efficiency, we employ a lightweight multilayer perceptron (MLP) combined with parallel triple-scoring for subgraph retrieval. To ensure effectiveness, we encode tailored structural distances from the query topic entities as structural features. This enables our MLP retriever to outperform more complex models, such as GNNs, LLMs, and heuristic searches, in terms of covering the triples and entities critical for answering the query while maintaining high efficiency. Additionally, the retrieved subgraphs have flexible forms, with adjustable sizes to accommodate the varying capacities of LLMs. SubgraphRAG employs unfine-tuned LLMs, maintaining generalization, adaptability to updated KGs, and compatibility with black-box LLMs.

We evaluate SubgraphRAG on two multi-hop knowledge graph question answering benchmarks—WebQSP and CWQ. Remarkably, without fine-tuning, smaller models like Llama3.1-8B-Instruct can achieve competitive performance. Larger models, such as GPT-4o, deliver state-of-the-art results. Furthermore, SubgraphRAG shows robust multi-hop reasoning capabilities, excelling on more complex, multi-hop questions and demonstrating effective generalization across datasets despite domain shifts. Ablation studies on different retrievers highlight the advantage of our retriever, which consistently outperforms baseline retrievers and is key to our superior KGQA performance. Additionally, our method also exhibits a substantial capability of reducing hallucination by generating knowledge-grounded answers and explanations for its reasoning.

## 2 Preliminaries

A KG can be represented as a set of triples, denoted by $\mathcal{G} = \{(h, r, t) \mid h, t \in \mathcal{E}, r \in \mathcal{R}\}$, where $\mathcal{E}$ represents the set of entities and $\mathcal{R}$ represents the set of relations. Each triple denoted by $\tau = (h, r, t)$ characterizes a fact that the head entity $h$ and the tail entity $t$ follow a directed relation $r$. In practice, entities and relations are often associated with a raw text surface form friendly for LLM reasoning.

**KG-based RAG** enhances LLMs with KG knowledge. Given a query $q$, LLMs can access relevant knowledge represented by triples in the KG to address $q$. The challenge lies in efficiently searching for relevant knowledge within the often large-scale KG and reasoning for accurate prediction.

**Entity Linking** is often the first step in KG-based RAG, whose goal is to identify the set of entities $\mathcal{T}_q \subset \mathcal{E}$ directly involved in the query $q$. The entities in $\mathcal{T}_q$, named topic entities, provide valuable inductive bias for retrieval as the triples relevant to $q$ are often close to $\mathcal{T}_q$.

**Knowledge Graph Question Answering** (KGQA) is an important application often used to evaluate KG-based RAG, where the query $q$ is a question that requires finding answers that satisfy certain constraints. The answer(s) $\mathcal{A}_q$ often correspond to a set of entities in the KG. Questions that require multiple triples in the KG as evidence to identify an answer entity are classified as complex questions, as they demand multi-hop reasoning, in contrast to single-hop questions.

## 3 The SubgraphRAG framework

Our proposed framework, SubgraphRAG, adopts a retrieval-and-reasoning pipeline for KG-based RAG. Specifically, given a query $q$, SubgraphRAG first extracts a subgraph $\mathcal{G}_q \subset \mathcal{G}$ that represents predicted relevant knowledge, and then LLMs generate the response by reasoning over $\mathcal{G}_q$. While some existing works adopt a similar pipeline (Zhang et al., 2022; Luo et al., 2024b; Wu et al., 2023; Wen et al., 2023), our approach is novel in addressing three key challenges: First, the extracted subgraph $\mathcal{G}_q$ is designed to comprehensively cover the relevant evidence for answering $q$, while adhering to a size constraint $K$, which can be adjusted according to the capacity of the downstream LLM. Second, the extraction of $\mathcal{G}_q$ is highly efficient and scalable. Third, we employ tailored prompting to guide the LLM in reasoning over $\mathcal{G}_q$ and generating a well-grounded answer with explanations.

### 3.1 Efficient, Flexible and Expressive Subgraph Retrieval

**Problem Reduction** We formulate the subgraph retrieval problem and reduce it to an efficiently solvable problem. An LLM models $\mathbb{P}(\cdot|\mathcal{G}_q, q)$ that takes queries and evidence subgraphs. Given a query $q$ and its answer $\mathcal{A}_q$, the best subgraph evidence for this LLM is denoted as $\mathcal{G}_q^* = \arg\max_{\mathcal{G}_q \subseteq \mathcal{G}} \mathbb{P}(\mathcal{A}_q \mid \mathcal{G}_q, q)$. However, solving this problem is impossible as it requires the knowledge of $\mathcal{A}_q$. Instead, we aim to learn a subgraph retriever that generalizes to unseen future queries.

Specifically, let the subgraph retriever be a distribution $\mathbb{Q}_\theta(\cdot|q, \mathcal{G})$ over the subgraph space of the KG, where $\theta$ denotes the parameters. Given a training set of question-answer pairs $\mathcal{D}$, the subgraph retriever learning problem can be formulated as the following problem:

$$\max_\theta \ \mathbb{E}_{(q,\mathcal{A}_q)\sim\mathcal{D}, \mathcal{G}_q \sim \mathbb{Q}_\theta(\mathcal{G}_q|q,\mathcal{G})} \ \mathbb{P}(\mathcal{A}_q \mid \mathcal{G}_q, q). \tag{1}$$

This problem is hard to solve due to the complexity of the LLM ($\mathbb{P}$). Evaluating $\mathbb{P}(\mathcal{A}_q \mid \mathcal{G}_q, q)$ requires the LLM to generate the particular answer $\mathcal{A}_q$, which can be costly and is only applicable to grey/white-box LLMs with accessible output logits, let alone the non-computable gradient $\frac{d\mathbb{P}}{d\mathcal{G}_q}$.

To solve the problem in Eq. 1, we adopt the following strategy. If we know the optimal subgraph $\mathcal{G}_q^*$, the maximum likelihood estimation (MLE) principle can be leveraged to train the retriever $\max_\theta \ \mathbb{E}_{(q,\mathcal{A}_q)\sim\mathcal{D}} \ \mathbb{Q}_\theta(\mathcal{G}_q^* \mid \mathcal{G}, q)$. However, getting $\mathcal{G}_q^*$ for an even known question-answer pair $(q, \mathcal{A}_q)$ is computationally hard and LLM-dependent. Instead, we use $(q, \mathcal{A}_q)$ to construct surrogate subgraph evidence with heuristics $\tilde{\mathcal{G}}(q, \mathcal{A}_q)$ and train the retriever based on MLE:

$$\max_\theta \ \mathbb{E}_{(q,\mathcal{A}_q)\sim\mathcal{D}} \ \mathbb{Q}_\theta(\tilde{\mathcal{G}}_q \mid \mathcal{G}, q), \quad \text{where } \tilde{\mathcal{G}}_q = \tilde{\mathcal{G}}(q, \mathcal{A}_q). \tag{2}$$

An example $\tilde{\mathcal{G}}_q$ is the shortest paths between the topic entities $\mathcal{T}_q$ and answer entities $\mathcal{A}_q$. Eq. 2 is conceptually similar to the weak supervision in Zhang et al. (2022). However, the formulation in Eq. 2 indicates that the sampled subgraph does not necessarily follow a fixed type like path. Instead, the retriever distribution $\mathbb{Q}_\theta$ can by construction factorize into a product of distributions over triples, allowing efficient training and inference, flexible subgraph forms, and adjustable subgraph sizes.

**Triple Factorization** We propose to adopt a retriever that allows a subgraph distribution factorization over triples given some latent variables $z_\tau = z_\tau(\mathcal{G}, q)$ (to be elaborated later):

$$\mathbb{Q}_\theta(\mathcal{G}_q \mid \mathcal{G}, q) = \prod_{\tau \in \mathcal{G}_q} p_\theta(\tau \mid z_\tau, q) \prod_{\tau \in \mathcal{G}\backslash\mathcal{G}_q} (1 - p_\theta(\tau \mid z_\tau, q)), \tag{3}$$

This strategy is inspired by the studies on graph generative models (Kipf & Welling, 2016) and enjoys four benefits: Efficiency in Training - The problem in Eq. 2 can be factorized as $\max_\theta \ \mathbb{E}_{(q,\mathcal{A}_q)\sim\mathcal{D}} \ \sum_{\tau \in \tilde{\mathcal{G}}_q} \log p_\theta(\tau \mid z_\tau, q) + \sum_{\tau \in \mathcal{G}\backslash\tilde{\mathcal{G}}_q} \log(1 - p_\theta(\tau \mid z_\tau, q))$; Efficiency in Sampling - After computing $z_\tau$, we can select triples $\tau$ from $\mathcal{G}$ in parallel; Flexibility - Triple combinations can form arbitrary subgraphs; Adjustable Size - Subgraphs formed by top-$K$ triples with different $K$ values can accommodate various LLMs with diverse reasoning capabilities. In practice, $\mathbb{Q}_\theta$ can be further simplified given topic entities $\mathcal{T}_q$ (He et al., 2021; Jiang et al., 2023b), by only considering subgraphs close to the topic entities, i.e., $p_\theta(\tau \mid z_\tau, q) = 0$ for a $\tau$ that is far from $\mathcal{T}_q$.

**Relevant Designs** Previous approaches often adopt heuristics and focus on particular subgraph types, such as constrained subgraph search (e.g., connected subgraphs (He et al., 2024)), constrained path search from topic entities, often imposing constraints in path counts and lengths (Luo et al., 2024b; Mavromatis & Karypis, 2024) or employing potentially expensive iterative processes (Zhang et al., 2022; Wu et al., 2023; Sun et al., 2024a; Liu et al., 2024a; Sun et al., 2024b), and entity selection followed by extracting entity-induced subgraphs, where all triples involving an entity are included together (Yasunaga et al., 2021; Taunk et al., 2023). The loss in flexibility narrows the space of possible retrieved subgraphs, which eventually harms the RAG effectiveness.

**Directional Distance Encoding (DDE)** $z_\tau(\mathcal{G}, q)$ models the relationship between a triple $\tau$ and the query $q$ given $\mathcal{G}$. Graph neural network (GNN) is one option for this modeling purpose through message passing with entity, relation, and question embeddings (Yasunaga et al., 2021; Kang et al., 2023; Mavromatis & Karypis, 2024; Liu et al., 2024a). However, it faces fundamental limitations in its representation power (Xu et al., 2019; Murphy et al., 2019; Morris et al., 2019; Chen et al., 2020).

> **System:** *Based on the triples retrieved from a knowledge graph, please answer the question. Please return formatted answers as a list, each prefixed with ``ans:''.*
>    **User:** Triplets: $(e_1, r_{12}, e_2)$ \n $(e_3, r_{34}, e_4)$ \n ... \n   Question: ...    *// ICL example*
> **Assistant:** *To answer the question, we have to find .... From the triples we can see that .... Therefore, the answers are: \n ans: ... \n ans: ... \n ....*   *// ICL example*
>
>    **User:** Triplets: $(e_a, r_{ab}, e_b)$ \n $(e_c, r_{cd}, e_d)$ \n ... \n   Question: ...   *// the evaluation question*
> **Assistant:** *To answer the question, we have to find .... From the triples we can see that .... Therefore, the answers are: \n ans: ... \n ans: ... \n ....*   *// the answer*

Figure 2: The prompt used in SubgraphRAG. Concrete examples can be found in Appendix E).

One critical limitation is its inability to compute distances between entities and topic entities (Li et al., 2020), which can be particularly problematic for multi-hop question answering.

The structural relationship between $\tau$ and $q$ provides valuable information complementing their semantic relationship. Inspired by the success of distance encoding and labeling trick in enhancing the structural representation power of GNNs (Li et al., 2020; Zhang et al., 2021), we propose a DDE as $z_\tau(\mathcal{G}, q)$ to model the structural relationship. Given topic entities $\mathcal{T}_q$, let $\mathbf{s}_e^{(0)}$ be a one-hot encoding representing whether $e \in \mathcal{T}_q$ or not. For the $l+1$-th round, we perform feature propagation and compute $\mathbf{s}_e^{(l+1)} = \text{MEAN}\{\mathbf{s}_{e'}^{(l)} \mid (e', \cdot, e) \in \mathcal{G}\}$, and through the reverse direction to account for the directed nature of $\mathcal{G}$, $\mathbf{s}_e^{(r,l+1)} = \text{MEAN}\{\mathbf{s}_{e'}^{(r,l)} \mid (e, \cdot, e') \in \mathcal{G}\}$, where $\mathbf{s}_e^{(r,0)} = \mathbf{s}_e^{(0)}$. We concatenate all results to obtain the final entity encodings $\mathbf{s}_e = [\mathbf{s}_e^{(0)} \| \mathbf{s}_e^{(1)} \| \cdots \| \mathbf{s}_e^{(r,1)} \| \cdots]$, leading to triple encodings $z_\tau(\mathcal{G}, q) = [\mathbf{s}_h \| \mathbf{s}_t]$ that concatenate the encodings of $h$ and $t$. In section 4.1, we compare different approaches to compute $z_\tau(\mathcal{G}, q)$ - using GNNs, DDE, or only one-hot encodings of $\mathcal{T}_q$, and DDEs perform the best. See Appendix A for additional illustrations and discussions.

**A Lightweight Implementation For** $p_\theta(\cdot \mid z_\tau(\mathcal{G}, q), q)$ We present a lightweight implementation of $p_\theta$ that integrates structural and semantic information. Following previous approaches (Karpukhin et al., 2020; Gao et al., 2024b), we employ off-the-shelf pre-trained text encoders to embed all entities/relations in a KG based on their text attributes. These semantic text embeddings are computed and stored in a vector database during the pre-processing stage for efficient retrieval. For a newly arrived question $q$, we embed $q$ to obtain $z_q$ and retrieve embeddings $z_h, z_r, z_t$ from the vector database for the involved entities and relations. After computing DDEs $z_\tau$, an MLP is employed for binary classification using the concatenated input $[z_q \| z_h \| z_r \| z_t \| z_\tau]$.

**Relevant Designs** We considered several alternatives but found them less preferable due to concerns about efficiency and adaptability to KG updates. Cross-encoders, which concatenate a question and a retrieval candidate for joint embedding (Wolf et al., 2019), potentially offer better retrieval performance. However, due to the inability to pre-compute embeddings, this approach significantly reduces retrieval efficiency when dealing with a large number of retrieval candidates, as is the case in triple retrieval. Li et al. (2023a) embeds each triple as a whole rather than individual entities and relations. However, this approach incurs higher computational and storage costs and exhibits reduced generalizability to the triples that are new combinations of old entities and relations. Our lightweight implementation allows for fast triple scoring while maintaining good generalizability.

## 3.2 PROMPTING-BASED LLM REASONING

We employ an LLM to reason over $\mathcal{G}_q$ by incorporating the linearized list of triples forming $\mathcal{G}_q$ into the prompt. This approach allows the LLM to ground its reasoning in the retrieved structured knowledge and select the answers $\hat{\mathcal{A}}_q$ from the entities within $\mathcal{G}_q$, mitigating issues like hallucinations and outdated knowledge (Lin et al., 2019; Shuster et al., 2021; Vu et al., 2024). Specifically, we prompt the LLM to generate knowledge-grounded explanations besides the answers, which further effectively reduces hallucinations. We adopt in-context learning (ICL) (Brown et al., 2020) and design dedicated prompt templates with explanation demonstrations to guide LLM reasoning (see Fig. 2).

Without fine-tuning LLMs, we reduce computational costs, enable the usage of state-of-the-art (SOTA) black-box LLMs, and maintain the generalizability for unseen KGs. Fine-tuning also risks enhancing prediction accuracy at the cost of general reasoning and explaining capabilities. High-quality text-based explanations are typically unavailable for question-answering tasks in practice. As a result, previous KG-based RAG approaches that rely on fine-tuning often depend on larger, unfine-tuned LLMs like GPT-4 to generate explanations as additional training data (Luo et al., 2024b).

Regarding the size $K$ of the retrieved subgraph, while increasing $K$ in principle improves the coverage of relevant information, it also risks introducing more irrelevant information that may harm

Table 1: Evaluation results for retrieval recall and wall-clock time. Best results are in **bold**. Being training-free, cosine similarity and G-Retriever stay unchanged in generalization evaluations.

| Model | WebQSP | | | | CWQ | | | | CWQ→WebQSP | | | WebQSP→CWQ | | |
|---|---|---|---|---|---|---|---|---|---|---|---|---|---|---|
| | Triples | | Entites | | Triples | | Entites | | Triples | | Entites | Triples | | Entites |
| | Shortest Path | GPT-4o | Answer | Time (s) | Shortest Path | GPT-4o | Answer | Time (s) | Shortest Path | GPT-4o | Answer | Shortest Path | GPT-4o | Answer |
| cosine similarity | 0.714 | 0.719 | 0.708 | **3** | 0.488 | 0.567 | 0.582 | 13 | 0.714 | 0.719 | 0.708 | 0.488 | 0.567 | 0.582 |
| SR+NSM w E2E | 0.487 | 0.504 | 0.707 | 101 | - | - | - | - | - | - | - | - | - | - |
| Retrieve-Rewrite-Answer | 0.058 | 0.062 | 0.740 | 69 | - | - | - | - | - | - | - | - | - | - |
| RoG | 0.713 | 0.388 | 0.807 | 948 | 0.623 | 0.298 | 0.841 | 2327 | 0.589 | 0.323 | 0.658 | 0.301 | 0.139 | 0.412 |
| G-Retriever | 0.294 | 0.325 | 0.545 | 672 | 0.183 | 0.217 | 0.375 | 1530 | 0.294 | 0.325 | 0.545 | 0.183 | 0.217 | 0.375 |
| GNN-RAG | 0.522 | 0.405 | 0.818 | 68 | 0.500 | 0.386 | 0.841 | 160 | 0.446 | 0.364 | 0.691 | 0.444 | 0.351 | 0.697 |
| SubgraphRAG | **0.883** | **0.865** | **0.944** | **6** | **0.811** | **0.840** | **0.914** | **12** | **0.794** | **0.776** | **0.887** | **0.622** | **0.623** | **0.773** |

Table 2: Breakdown of recall evaluation over # hops. Best results are in **bold**.

| Model | Shortest Path Triple Recall | | | | | GPT-4o Triple Recall | | | | | Answer Entity Recall | | | | |
|---|---|---|---|---|---|---|---|---|---|---|---|---|---|---|---|
| | WebQSP | | CWQ | | | WebQSP | | CWQ | | | WebQSP | | CWQ | | |
| | 1 (65.8%) | 2 (34.2%) | 1 (28.0%) | 2 (65.9%) | ≥ 3 (6.1%) | 1 (65.8%) | 2 (34.2%) | 1 (28.0%) | 2 (65.9%) | ≥ 3 (6.1%) | 1 (65.8%) | 2 (34.2%) | 1 (28.0%) | 2 (65.9%) | ≥ 3 (6.1%) |
| cosine similarity | 0.874 | 0.405 | 0.629 | 0.442 | 0.333 | 0.847 | 0.483 | 0.629 | 0.511 | 0.464 | 0.943 | 0.253 | 0.903 | 0.472 | 0.289 |
| SR+NSM w E2E | 0.565 | 0.324 | - | - | - | 0.580 | 0.376 | - | - | - | 0.916 | 0.301 | - | - | - |
| Retrieve-Rewrite-Answer | 0.064 | 0.046 | - | - | - | 0.062 | 0.061 | - | - | - | 0.745 | 0.729 | - | - | - |
| RoG | 0.869 | 0.415 | 0.766 | 0.597 | 0.253 | 0.446 | 0.271 | 0.347 | 0.293 | 0.122 | 0.874 | 0.677 | 0.920 | 0.827 | 0.628 |
| G-Retriever | 0.335 | 0.216 | 0.134 | 0.205 | 0.168 | 0.345 | 0.284 | 0.159 | 0.240 | 0.226 | 0.596 | 0.446 | 0.377 | 0.384 | 0.269 |
| GNN-RAG | 0.532 | 0.502 | 0.515 | 0.498 | 0.446 | 0.384 | 0.445 | 0.328 | 0.408 | 0.418 | 0.810 | 0.831 | 0.853 | 0.841 | 0.787 |
| MLP | 0.828 | 0.687 | 0.651 | 0.690 | 0.534 | 0.811 | 0.781 | 0.635 | 0.707 | 0.616 | 0.933 | 0.874 | 0.932 | 0.870 | **0.793** |
| MLP + topic entity | 0.944 | 0.729 | **0.854** | 0.750 | 0.560 | 0.884 | 0.775 | 0.769 | 0.773 | 0.647 | 0.976 | 0.843 | **0.956** | 0.885 | 0.665 |
| SubgraphRAG | **0.953** | **0.748** | 0.831 | **0.820** | **0.626** | **0.908** | **0.809** | **0.823** | **0.860** | **0.755** | **0.977** | **0.881** | 0.946 | **0.916** | 0.741 |

LLM reasoning (Xu et al., 2024; Liu et al., 2024b). Different LLMs are inherently equipped with different sized context windows and also exhibit distinct capabilities in reasoning over long-context retrieval results (Dubey et al., 2024). As such, although the training of SubgraphRAG retriever is LLM-agnostic, the size $K$ needs to be properly selected per LLM and constraints. In Section 4.2, we empirically verify that more powerful LLMs can benefit from incorporating a larger-sized retrieved subgraph, demonstrating the benefit of size-adjustable subgraph retrieval in SubgraphRAG.

## 4 EXPERIMENTS

We design our empirical studies to examine the effectiveness and efficiency of SubgraphRAG in addressing the various challenges inherent to KG-based RAG, covering both retrieval and reasoning aspects. **Q1**) Overall, to meet the accuracy and low-latency requirements of KG-based RAG, does SubgraphRAG effectively and efficiently retrieve relevant information? **Q2**) For complex questions involving multi-hop reasoning and multiple topic entities, does SubgraphRAG properly integrate structural information for effective retrieval? **Q3**) How effectively does SubgraphRAG perform on KGQA tasks, and how is its accuracy influenced by different factors? **Q4**) To what extent can our pipeline provide effective knowledge-grounded explanations for question answering?

**Datasets.** We adopt two prominent and challenging KGQA benchmarks that necessitate multi-hop reasoning – WebQSP (Yih et al., 2016) and CWQ (Talmor & Berant, 2018). Both benchmarks utilize Freebase (Bollacker et al., 2008) as the underlying KG. To evaluate the capability of LLM reasoners in knowledge-grounded hallucination-free question answering, we introduce WebQSP-sub and CWQ-sub, where we remove samples whose answer entities are absent from the KG.

### 4.1 EVALUATION FOR RETRIEVAL (**Q1** & **Q2**)

**Baseline Retrievers.** **Cosine similarity** performs structure-free triple retrieval based on triple embeddings (Li et al., 2023a). **SR+NSM w E2E** (Zhang et al., 2022) and **Retrieve-Rewrite-Answer** (Wu et al., 2023) propose constrained path search based on pre-trained text encoders. **RoG** (Luo et al., 2024b) adopts a similar strategy but fine-tunes an LLM for relation path prediction. **G-Retriever** (He et al., 2024) combines cosine similarity search with combinatorial optimization to construct a connected subgraph. **GNN-RAG** employs lightweight GNNs to predict answer entities and extracts shortest paths between them and topic entities (Mavromatis & Karypis, 2022; 2024).

**Implementation Details.** We adopt gte-large-en-v1.5 (Li et al., 2023c) as the pre-trained text encoder for both the cosine similarity baseline and SubgraphRAG, a 434M model that achieves a good balance between efficiency and English retrieval performance, as evidenced by the Massive Text Embedding Benchmark (MTEB) leaderboard (Muennighoff et al., 2023). For supervision signals, there are no ground-truth relevant subgraphs for a query $q$. Previous path-based subgraph retrievers adopt the shortest paths between the topic and answer entities for the weak supervision signals (Zhang et al., 2022; Luo et al., 2024b). We also utilize these shortest paths as the heuristic relevant sub-

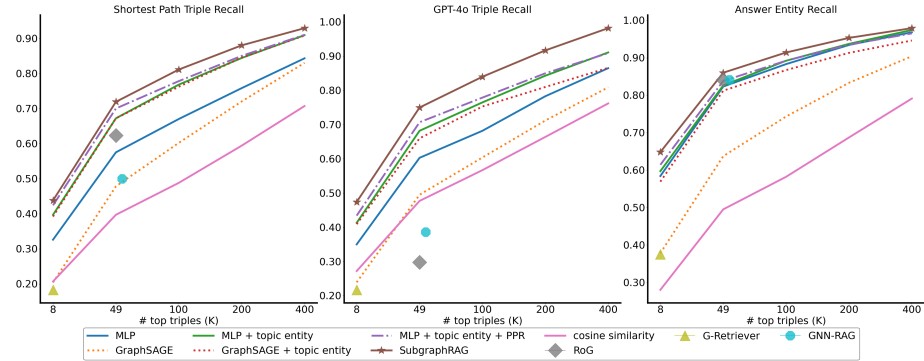

Figure 3: Retrieval effectiveness on CWQ across a spectrum of $K$ values for top-$K$ triple retrieval.

graphs $\tilde{\mathcal{G}}_q$ to train $\mathbb{Q}_\theta$ as in Eq. 2. To reduce the size of candidate triples $\mathcal{G}$, we construct subgraphs centered at topic entities $\mathcal{T}_q$, following previous works. See Appendix C.1 for more details.

**Evaluation Metrics.** Our evaluation covers both effectiveness and efficiency. For effectiveness, we employ three recall metrics, respectively for shortest path triples (i.e., $\tilde{\mathcal{G}}_q$), GPT-4o-identified relevant triples, and answer entities within retrieved subgraphs. The first metric assesses the retrieval of weak supervision signals used for training. For a more accurate assessment, we employ GPT-4o to identify $\leq 20$ relevant triples for all test samples, potentially capturing relevant triples beyond the shortest paths (see Appendix E for labeling details). We calculate individual recall values per question and then take the average. For efficiency, we measure wall clock time on a 48GB NVIDIA RTX 6000 Ada GPU in seconds. To focus on model computational efficiency, we exclude KG query time. For text embedding, employed by SubgraphRAG variants and most baselines (cosine similarity, G-Retriever, and GNN-RAG), we only count the time for question embedding as the entity and relation embeddings for the entire KG can be pre-computed. Such pre-computation may yield a time-memory trade-off.

**Overall Evaluation (Q1).** Table 1 shows that SubgraphRAG consistently outperforms all other approaches in retrieval effectiveness.

Table 3: Question-answering performance on WebQSP and CWQ. Best results are in **bold**. By default, our reasoners use the top 100 retrieved triples. Results with 200 and 500 triples (indicated in parentheses) are also shown. Results with ($\leftrightarrow$) evaluate retriever generalizability, where the retriever is trained on one dataset and applied to the other.

| | WebQSP | | CWQ | |
|---|---|---|---|---|
| | Macro-F1 | Hit | Macro-F1 | Hit |
| UniKGQA | 72.2 | - | 49.0 | - |
| SR+NSM w E2E | 64.1 | - | 46.3 | - |
| KD-CoT | 52.5 | 68.6 | - | 55.7 |
| ToG (GPT-4) | - | 82.6 | - | **67.6** |
| StructGPT | - | 74.69 | - | - |
| Retrieve-Rewrite-Answer | - | 79.36 | - | - |
| G-Retriever | 53.41 | 73.46 | - | - |
| RoG-Joint | 70.26 | 86.67 | 54.63 | 61.94 |
| RoG-Sep | 66.45 | 82.19 | 53.87 | 60.55 |
| EtD | - | 82.5 | - | 62.0 |
| GNN-RAG | 71.3 | 85.7 | 59.4 | 66.8 |
| SubgraphRAG + Llama3.1-8B | 70.57 | 86.61 | 47.16 | 56.98 |
| SubgraphRAG + Llama3.1-70B | 74.70 | 86.24 | 51.78 | 57.89 |
| SubgraphRAG + ChatGPT | 69.21 | 83.11 | 49.13 | 56.27 |
| SubgraphRAG + GPT4o-mini | 77.45 | 90.11 | 54.13 | 62.02 |
| SubgraphRAG + GPT4o | 76.46 | 89.80 | 59.08 | 66.69 |
| SubgraphRAG + GPT4o-mini (200) | 77.82 | 90.54 | 54.69 | 63.49 |
| SubgraphRAG + GPT4o (200) | 78.24 | 90.91 | 59.42 | 67.49 |
| SubgraphRAG + GPT4o-mini (500) | 77.67 | **91.22** | 55.41 | 64.97 |
| SubgraphRAG + Llama3.1-8B ($\leftrightarrow$) | 66.42 | 83.42 | 37.96 | 48.57 |
| SubgraphRAG + GPT4o-mini ($\leftrightarrow$) | 73.81 | 88.08 | 44.69 | 54.21 |
| SubgraphRAG + GPT4o-mini ($\leftrightarrow$, 500) | 76.20 | 91.22 | 50.30 | 60.80 |
| SubgraphRAG + GPT4o ($\leftrightarrow$) | 74.27 | 87.10 | 47.55 | 54.89 |
| SubgraphRAG + GPT4o ($\leftrightarrow$, 500) | 77.12 | 89.68 | 55.14 | 62.98 |

RoG achieves the most competitive baseline performance for retrieving shortest path triples. However, when evaluated on GPT-4o-labeled triples, RoG's performance drops significantly (45.6% for WebQSP, 52.2% for CWQ), while SubgraphRAG remains robust (2.0% decrease for WebQSP, 3.6% increase for CWQ). This stark difference, despite using the same training signals, empirically validates that SubgraphRAG's individual triple selection mechanism allows for more flexible and effective subgraph extraction compared to RoG's constrained path search approach. Regarding **efficiency**, SubgraphRAG is only slightly slower than the cosine similarity baseline on WebQSP while being one to two orders of magnitude faster than other baselines, including GNN-RAG. For the cosine similarity baseline and SubgraphRAG, we report recall metrics based on the top-100 retrieved triples (2.3% of total candidate triples on average). This budget consistently yields robust reasoning performance across LLMs in subsequent experiments.

**Generalizability.** We further examine the generalizability of the retrievers by training them on dataset $A$ and evaluating on dataset $B$, denoted as $A \rightarrow B$ in Table 1. Despite an anticipated performance degradation, SubgraphRAG consistently outperforms the alternative approaches.

**Ablation Study for Design Options and Retrieval Size.** To evaluate individual component contributions, we conduct an ablation study with several variants. **MLP** is a structure-free variant employing only text embeddings. We consider **GraphSAGE** (Hamilton et al., 2017), a popular

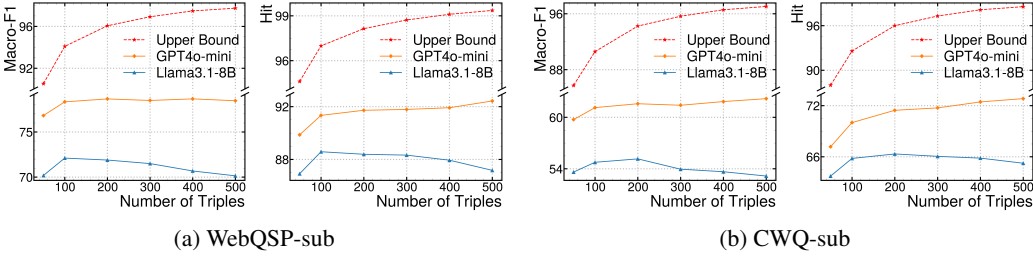

(a) WebQSP-sub  (b) CWQ-sub

Figure 4: Ablation studies on the number of retrieved triples used in LLM reasoners.

GNN, to update entity representations prior to the triple scoring. We further augment both MLP and GraphSAGE with a one-hot-encoding topic entity indicator (**MLP + topic entity** and **GraphSAGE + topic entity**). Following Sun et al. (2018), we incorporate Personalized PageRank (PPR) (Haveliwala, 2002), seeded from the topic entities, to integrate structural information (**MLP + topic entity + PPR**). To account for both the varying capabilities of downstream LLMs and inference cost/latency constraints, we evaluate these variants across a broad spectrum of retrieval sizes ($K$).

Fig. 3 presents the results on CWQ, with baselines included for reference. Larger retrieval sizes uniformly improve recall across all variants. Equipped with DDE, SubgraphRAG outperforms other variants, even at the relatively small average retrieval sizes of the baselines. This demonstrates that SubgraphRAG's superiority is not solely attributable to larger retrieval sizes. Regarding design options, the topic entity indicator invariably leads to an improvement. In contrast, GNN variants often result in performance degradation compared to their MLP counterparts. We suspect that the diffusion of semantic information introduces noise in triple selection. Finally, PPR fails to reliably yield improvements. For the results on WebQSP, see Appendix C.3, which are also consistent.

**Multi-Hop and Multi-Topic Questions (Q2).** To evaluate the effectiveness of various approaches in capturing structural information for complex multi-hop and multi-topic questions, we group questions based on the number of hops and topic entities. Table 2 presents the performance breakdown by hop count. SubgraphRAG consistently outperforms other methods on WebQSP and achieves the best overall performance on CWQ. Notably, while the cosine similarity baseline and RoG demonstrate competitive performance for single-hop questions, their performance degrades significantly for multi-hop questions. Appendix C.4 provides a performance breakdown for single-topic and multi-topic questions, focusing exclusively on CWQ due to the predominance of single-topic questions in the WebQSP test set (98.3%). SubgraphRAG consistently exhibits superior performance across all metrics for both single-topic and multi-topic questions. Our comprehensive analysis highlights the remarkable effectiveness of DDE in capturing complex topic-centered structural information essential for challenging questions involving multi-hop reasoning and multiple topic entities.

## 4.2 KGQA RESULTS (**Q3** & **Q4**)

**KGQA Baselines.** Besides the baselines in retrieval evaluation, we include more KGQA methods, including UniKGQA (Jiang et al., 2023b), KD-CoT (Wang et al., 2023a), StructGPT (Jiang et al., 2023a), ToG (Sun et al., 2024a), and EtD (Liu et al., 2024a). For RoG, we present two entries: RoG-Joint and RoG-Sep. Originally, RoG fine-tuned its LLMs on the training sets of both WebQSP and CWQ. Yet, this joint training approach leads to significant label leakage, with over 50% of WebQSP test questions (or their variants) appearing in CWQ's training set, and vice versa. Therefore, we re-trained RoG on each dataset separately, denoted by RoG-Sep.

**Evaluation Metrics.** Along with the commonly reported Macro-F1 and Hit[1], we also include Micro-F1 to account for the imbalance in the number of ground-truth answers across samples and Hit@1 for a more inclusive evaluation. To further assess model performance, we introduce $score_h$, inspired by (Yang et al., 2024), which evaluates how truth-grounded the predicted answers are and the degree of hallucination. This metric penalizes hallucinated answers while favoring missing answers over incorrect ones. Scores are normalized to a range of 0 to 100, with higher scores indicating better truth-grounding in the model's answers. The scoring strategy is described in detail in Appendix D.

**Experiment Settings.** We use the vllm (Kwon et al., 2023) framework for efficient LLM inference. For UniKGQA, KD-CoT, Retrieve-Rewrite-Answer, ToG, and EtD, we reference their published

---

[1]The baselines claim to report Hit@1, but they actually compute Hit, which measures whether at least one correct answer appears in the LLM response.

Table 4: Question-answering performance on WebQSP-sub and CWQ-sub. Best results are in **bold**. By default, our reasoners use the top 100 retrieved triples. Results with 200 and 500 triples (indicated in parentheses) are also shown. Results with ($\leftrightarrow$) evaluate retriever generalizability, where the retriever is trained on one dataset and applied to the other.

| | WebQSP-sub | | | | | CWQ-sub | | | | |
|---|---|---|---|---|---|---|---|---|---|---|
| | Macro-F1 | Micro-F1 | Hit | Hit@1 | $Score_h$ | Macro-F1 | Micro-F1 | Hit | Hit@1 | $Score_h$ |
| SR+NSM w E2E | 58.79 | 37.04 | 68.63 | 60.62 | 64.44 | - | - | - | - | - |
| G-Retriever | 54.13 | 23.84 | 74.52 | 67.56 | 67.97 | - | - | - | - | - |
| RoG-Joint | 72.01 | 47.70 | 88.90 | 82.62 | 76.13 | 58.61 | 52.12 | 66.22 | 61.17 | 55.15 |
| RoG-Sep | 67.94 | 43.10 | 84.03 | 77.61 | 72.79 | 57.69 | 52.83 | 64.64 | 60.64 | 54.51 |
| SubgraphRAG + Llama3.1-8B | 72.10 | 46.56 | 88.58 | 84.80 | 82.42 | 54.76 | 51.76 | 65.80 | 59.69 | 62.89 |
| SubgraphRAG + Llama3.1-70B | 75.97 | 51.64 | 87.88 | 85.89 | **85.57** | 61.49 | 59.91 | 68.43 | 65.52 | __67.62__ |
| SubgraphRAG + ChatGPT | 70.81 | 44.73 | 85.18 | 80.82 | 81.53 | 56.37 | 54.44 | 64.40 | 60.99 | 61.31 |
| SubgraphRAG + GPT4o-mini | 78.34 | 58.44 | 91.34 | 87.36 | 82.21 | 61.13 | 58.86 | 70.01 | 65.48 | 64.20 |
| SubgraphRAG + GPT4o | 77.61 | 56.78 | 91.40 | 86.40 | 81.85 | 65.99 | __63.18__ | 73.91 | 68.89 | 66.57 |
| SubgraphRAG + GPT4o-mini (200) | 78.66 | 58.65 | 91.73 | 87.04 | 81.98 | 61.58 | 57.47 | 71.45 | 65.87 | 63.66 |
| SubgraphRAG + GPT4o (200) | __79.40__ | __58.91__ | __92.43__ | 87.75 | 82.46 | __66.48__ | 61.30 | __75.14__ | __69.42__ | 66.45 |
| SubgraphRAG + GPT4o-mini (500) | 78.46 | 57.08 | __92.43__ | __88.01__ | 81.95 | 62.18 | 56.86 | 72.82 | 66.57 | 62.77 |
| SubgraphRAG + Llama3.1-8B ($\leftrightarrow$) | 67.91 | 42.79 | 85.25 | 81.21 | 80.09 | 43.03 | 40.73 | 55.09 | 47.58 | 56.78 |
| SubgraphRAG + GPT4o-mini ($\leftrightarrow$) | 74.42 | 49.41 | 89.10 | 84.67 | 81.35 | 49.47 | 45.16 | 60.18 | 54.18 | 58.86 |
| SubgraphRAG + GPT4o-mini ($\leftrightarrow$, 500) | 76.83 | 52.01 | 92.30 | 87.43 | 81.41 | 55.58 | 49.13 | 67.24 | 59.69 | 59.87 |
| SubgraphRAG + GPT4o ($\leftrightarrow$) | 74.98 | 50.67 | 88.26 | 84.73 | 83.24 | 52.40 | 52.28 | 60.36 | 56.39 | 63.70 |
| SubgraphRAG + GPT4o ($\leftrightarrow$, 500) | 77.96 | 56.40 | 90.96 | 86.66 | 83.35 | 61.96 | 58.48 | 70.47 | 64.96 | 66.86 |

Table 5: Breakdown of QA performance by reasoning hops.

| | WebQSP-sub | | | | CWQ-sub | | | | | |
|---|---|---|---|---|---|---|---|---|---|---|
| | 1 (65.8%) | | 2 (34.2%) | | 1 (28.0%) | | 2 (65.9%) | | $\geq 3$ (6.1%) | |
| | Marco-F1 | Hit | Marco-F1 | Hit | Marco-F1 | Hit | Marco-F1 | Hit | Marco-F1 | Hit |
| G-Retriever | 56.41 | 78.20 | 45.73 | 65.35 | - | - | - | - | - | - |
| RoG-Joint | 77.05 | 92.96 | 62.53 | 81.54 | 59.75 | 66.33 | 59.70 | 68.56 | 41.46 | 43.27 |
| RoG-Sep | 74.50 | 89.83 | 55.62 | 73.45 | 59.35 | 66.20 | 59.45 | 67.17 | 31.33 | 33.33 |
| SubgraphRAG (Llama3.1-8B) | 75.50 | 91.40 | 65.87 | 83.62 | 51.54 | 63.05 | 57.52 | 68.93 | 41.88 | 47.37 |
| SubgraphRAG (GPT4o-mini) | 80.56 | 92.86 | 74.11 | 88.51 | 57.36 | 67.34 | 63.85 | 72.74 | 51.14 | 54.39 |

results due to various reproducibility difficulties. For StructGPT, we directly use their provided processed files to obtain results for WebQSP. For SR, we include their reported results for WebQSP and CWQ, and further evaluate it on WebQSP-sub as we could not run through SR's code on the two CWQ datasets. For the remaining baselines, we successfully reproduced their results and evaluated them on additional metrics and datasets, including WebQSP-sub and CWQ-sub. However, we do not include results for G-Retriever on CWQ as it required over 200 hours of computation on 2 NVIDIA RTX 6000 Ada GPUs. If not specified, the LLM reasoners in SubgraphRAG use the top 100 retrieved triples; results using more triples are explicitly noted. All Llama models used are their instruction fine-tuned versions, i.e., Llama3.1-8B(70B)-Instruct. The used GPT models' versions are gpt-3.5-turbo-1106, gpt-4o-mini-2024-07-18, and gpt-4o-2024-08-06. For all LLM reasoners used in SubgraphRAG, both the temperature and the seed are set to 0 to ensure reproducibility.

**Overall Performance.** Tables 3 and 4 present the results, where SubgraphRAG achieves SOTA on WebQSP and WebQSP-sub. Even with 8B LLMs, our method surpasses previous SOTA approaches by $\leq 4\%$ in Macro-F1 and Hit metrics (excluding RoG-Joint due to test label leakage). With larger models like Llama3.1-70B-Instruct and GPT-4o, SubgraphRAG achieves greater performance, showing $\leq 12\%$ improvement in Macro-F1 and $\leq 9\%$ in Hit. On the more challenging CWQ and CWQ-sub datasets, which require more reasoning hops, SubgraphRAG performs competitively with smaller 8B models. When paired with advanced LLMs like GPT-4o, SubgraphRAG achieves results second only to ToG on CWQ. Notably, SubgraphRAG requires only a single call to GPT-4o, whereas ToG requires 6-8 calls, which increases computational cost, and we were unable to reproduce ToG's performance using their published code. On CWQ-sub, SubgraphRAG demonstrates gains of $\leq 9\%$ in Macro-F1 and $\leq 11\%$ in Hit, indicating that tasks with greater reasoning complexity benefit substantially from more powerful LLMs. Our method also excels in truth-grounded QA, with $Score_h$ consistently outperforming baselines by $\leq 12\%$, driven by our prompt design that encourages retrieval-grounded reasoning. See Appendix D for further discussions.

Additionally, our framework generalizes well, with retrievers trained on one dataset performing effectively on others. Although moderate performance decay occurs due to domain shift, this can largely be mitigated by including more triples extracted by the retriever. Notably, the decay is generally minor on WebQSP and WebQSP-sub but more pronounced on CWQ and CWQ-sub, potentially due to greater label leakage from the WebQSP test set to the CWQ training set.

Table 6: Ablation studies with different retrievers, using the same prompt and Llama3.1-8B-Instruct as the reasoner. Rand refers to random triple sampling, RandNoAns removes triples with ground-truth answers after random sampling, and NoRetriever directly asks questions without KG info.

| | WebQSP | | CWQ | | WebQSP-sub | | | | | CWQ-sub | | | | |
|---|---|---|---|---|---|---|---|---|---|---|---|---|---|---|
| | Macro-F1 | Hit | Macro-F1 | Hit | Macro-F1 | Micro-F1 | Hit | Hit@1 | $Score_h$ | Macro-F1 | Micro-F1 | Hit | Hit@1 | $Score_h$ |
| Rand + SubgraphRAG Reasoner | 37.69 | 60.14 | 27.34 | 35.85 | 37.79 | 17.79 | 60.74 | 54.97 | 65.06 | 29.97 | 29.15 | 39.15 | 35.11 | 52.45 |
| RandNoAns + SubgraphRAG Reasoner | 21.18 | 33.54 | 16.40 | 22.71 | 20.61 | 8.64 | 33.03 | 27.33 | 47.29 | 16.47 | 16.44 | 23.00 | 19.42 | 43.79 |
| NoRetriever + SubgraphRAG Reasoner | 35.86 | 51.90 | 25.64 | 32.34 | 35.03 | 17.01 | 51.38 | 47.59 | 55.57 | 27.42 | 22.66 | 33.95 | 30.65 | 44.87 |
| cosine similarity + SubgraphRAG Reasoner | 58.41 | 74.14 | 34.59 | 43.61 | 59.26 | 37.31 | 75.43 | 71.20 | 73.16 | 39.05 | 35.48 | 49.02 | 43.68 | 55.72 |
| Retrieve-Rewrite-Answer + SubgraphRAG Reasoner | 8.96 | 11.43 | - | - | 9.11 | 5.23 | 11.43 | 10.84 | 63.45 | - | - | - | - | - |
| StructGPT + SubgraphRAG Reasoner | 62.14 | 75.00 | - | - | 62.55 | 44.72 | 75.69 | 73.57 | 80.82 | - | - | - | - | - |
| G-Retriever + SubgraphRAG Reasoner | 48.91 | 64.50 | 28.47 | 34.58 | 49.92 | 28.08 | 65.88 | 62.60 | 72.54 | 31.55 | 32.22 | 38.17 | 35.22 | 58.22 |
| RoG-Sep + SubgraphRAG Reasoner | 57.68 | 74.39 | 36.85 | 45.23 | 59.36 | 40.32 | 76.65 | 72.61 | 79.69 | 44.11 | 43.69 | 54.04 | 47.68 | 66.18 |
| GNN-RAG + SubgraphRAG Reasoner | 62.34 | 78.81 | 46.40 | 54.35 | 63.93 | 42.43 | 80.89 | 77.10 | 82.86 | 55.08 | 53.38 | 64.33 | 59.76 | 67.55 |
| GNN-RAG (↔) + SubgraphRAG Reasoner | 52.07 | 71.38 | 34.88 | 43.13 | 53.01 | 34.58 | 72.93 | 67.99 | 73.37 | 39.98 | 34.60 | 49.54 | 44.91 | 57.42 |
| SubgraphRAG | 70.57 | 86.61 | 47.16 | 56.98 | 72.10 | 46.56 | 88.58 | 84.80 | 82.42 | 54.76 | 51.76 | 65.80 | 59.69 | 62.89 |
| SubgraphRAG (↔) | 66.42 | 83.42 | 37.96 | 48.57 | 67.91 | 42.79 | 85.25 | 81.21 | 80.09 | 43.03 | 40.73 | 55.09 | 47.58 | 56.78 |

For efficiency, studies show that overall LLM latency is dominated by # LLM calls, followed by # output tokens and then # input tokens [2]. SubgraphRAG, despite potentially varying # input tokens, utilizes only one LLM call per question, whereas various baselines require multiple calls.

**Performance Breakdown & Analysis.** Table 5 presents the performance breakdown by reasoning hops. Our method, even with an 8B LLM, shows significantly better results on multi-hop questions. This can be attributed to our design philosophy, which does not restrict the retrieved subgraph form and leverages unfine-tuned LLMs with ICL examples, fully unlocking their reasoning capabilities. Interestingly, SubgraphRAG shows relatively lower performance on some 1-hop questions in CWQ-sub. Cross-referencing with Table 8, which provides a detailed breakdown of whether the answers are truth-grounded, we find that previous methods may generate many correct answers that are not supported by the retrieved results. Further investigation suggests this discrepancy may partly result from dataset leakage, giving tuning-based methods an advantage: In CWQ, over 60% of test samples have answers that have appeared in the training set, making it easier for these methods to succeed without robust reasoning. Thus, future studies are needed to fully investigate the potential leakage.

**Performance with Different Retrievers.** To assess the impact of our retriever on the overall performance, we conducted extensive studies by replacing the retrieval results with those from baseline retrievers, while keeping all other settings constant (same prompts, same LLM reasoners). Tables 6 and 9 show the results using Llama3.1 8B and GPT4o-mini as the LLM reasoners, respectively. For retrievers from Retrieve-Rewrite-Answer, StructGPT, G-Retriever, and RoG, we used all their retrieved triples, while for other cases, we kept 100 triples. Clearly, pairing with our retriever yields significantly better performance on all metrics and datasets compared to using baseline retrievers, indicating that our superior results are largely due to the effectiveness of our improved retriever.

**Impact of Retrieval Size.** Irrelevant context can mislead LLMs (Wu et al., 2024), and retrieving more triples may not help. We conducted experiments (Fig. 4) by feeding Llama 3.1 8B and GPT-4o-mini different numbers of retrieved triples. The upper bound is defined such that if the answer entity is present in the retrieved results, it is considered a correct answer. The results show a significant performance decay in Llama 3.1 8B with more triples, while GPT-4o-mini performs even better with additional retrieved triples. This suggests that different LLMs have varying capacities for robust reasoning, and users should input proper numbers of retrieved results to match the capacity of the used LLM and the available computational budget. See Appendix D for more results.

**Explainability.** By retrieving flexible, high-quality evidence subgraphs as context and exploiting the strong reasoning capabilities of pre-trained LLMs, SubgraphRAG can natively provide effective explanations along with answer predictions. In contrast, explainable predictions with fine-tuned LLMs necessitate extra labeling efforts for preserving explainability (Luo et al., 2024b) or post hoc explainability approaches (Krishna et al., 2023). To illustrate the explainability of our framework, we provide multiple examples of our reasoner's responses (see Appendix F).

## 5 CONCLUSION

We propose SubgraphRAG, a KG-based RAG framework that performs efficient and flexible subgraph retrieval followed by prompting unfine-tuned LLMs. SubgraphRAG demonstrates better or comparable accuracy, efficiency, and explainability compared to existing approaches.

---

[2]For LLMs like GPT-4 and Claude-3.5, an additional input token typically adds about $10^{-2}\times$ the latency of an extra output token and $10^{-4}\times$ that of an additional LLM call (Vivek, 2024).

ACKNOWLEDGMENTS

M. Li, S. Miao, and P. Li are partially supported by NSF awards PHY-2117997, IIS-2239565, IIS-2428777, and CCF-2402816; DOE award DE-FOA-0002785; JPMC faculty awards; OpenAI Researcher Access Program Credits; and Microsoft Azure Research Credits for Generative AI.

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

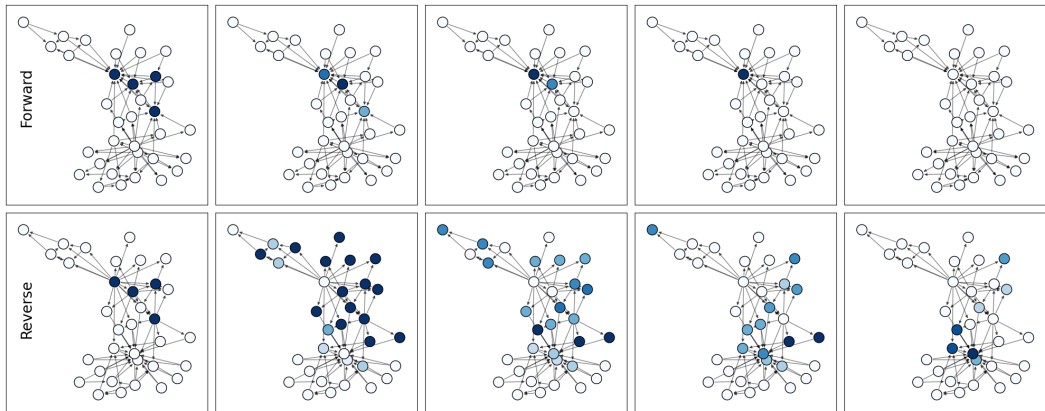

Figure 5: A visual illustration of DDE, where the leftmost column highlights the topic entities in color. From left to right, we iteratively perform graph diffusion over the directed edges. The color intensity indicates the magnitude of the diffusion weights for individual entities.

# A  DIRECTIONAL DISTANCE ENCODING (DDE)

Fig. 5 provides a visual illustration of DDE. By passing entity labeling information in both directions for multiple rounds, DDE efficiently approximates measures such as the average distance between an entity and all topic entities. This process implicitly reveals information like whether an entity lies at the intersection of paths of particular lengths connecting multiple topic entities.

# B  ADDITIONAL RELATED WORK

The field of KGQA has evolved significantly over time. Early approaches to KGQA do not rely on LLMs for answer generation (Yasunaga et al., 2021; Taunk et al., 2023; Zhang et al., 2022; Lin et al., 2019; Sun et al., 2019), though they often employ pre-trained language models (PLMs) like BERT as text encoders (Devlin et al., 2019). These methods typically search for answer entities with GNNs Yasunaga et al. (2021); Taunk et al. (2023); Lin et al. (2019) or LSTM-based models Sun et al. (2019); Zhang et al. (2022).

With the rapid advancement of LLMs in recent years, researchers began to leverage them in KGQA. For instance, recent work has explored using LLMs as translators, converting natural language questions into executable SPARQL queries for KG databases to retrieve the answers (Jiang et al., 2023a; Wang et al., 2023b; Luo et al., 2024a; Xiong et al., 2024).

Contemporary approaches have further expanded the role of LLMs, utilizing them for both knowledge retrieval from KGs and reasoning (Kim et al.; Gao et al., 2024a; Wang et al., 2024; Guo et al., 2024; Ma et al., 2024; Sun et al., 2024a; Jiang et al., 2024; Jin et al., 2024). The strength of this strategy lies in LLMs' ability to handle multi-hop tasks by breaking them down into manageable steps. However, as discussed in Section 1, this often necessitates multiple LLM calls, resulting in high latency. To mitigate this issue, some frameworks have attempted to fine-tune LLMs to memorize knowledge, but this reduces their ability to generalize to dynamically updated or novel KGs (Luo et al., 2024b; Mavromatis & Karypis, 2024). Other models have explored fine-tuning adapters embedded in fixed LLMs to better preserve their general reasoning capabilities while adapting to specific KGs (He et al., 2024; Gao et al., 2024a; Hu et al., 2024).

In parallel with these developments, several approaches have emerged that, like our approach, allow LLMs to reason over subgraphs (Kim et al.; Liu et al., 2024a; Li et al., 2023a; Guo et al., 2024; Wu et al., 2023; Li et al., 2024; Wen et al., 2023), though they employ different retrieval strategies. Kim et al. breaks queries into sub-queries, retrieving evidence for each sub-query before reasoning over the collected evidence. Guo et al. (2024) uses PLM-based entity extraction followed by multi-hop expansion for retrieval. Li et al. (2023a) linearizes KG triples into natural language for global triple retrieval using BM25 or dense passage retrievers. Wen et al. (2023) extracts topic entities and

merges triples into the retrieved subgraph using two heuristic methods: connecting topic entities via paths and retrieving their 1-hop neighbors.

While these subgraph retrieval strategies share similarities with SubgraphRAG, they lack several key advantages of it, such as retrieval efficiency, adjustable subgraph sizes, and flexible subgraph types. Consequently, they frequently result in suboptimal coverage of relevant information in the retrieved subgraphs. SubgraphRAG addresses these limitations, offering a more comprehensive and adaptable approach to KGQA that builds upon and extends the capabilities of existing methods, while fully leveraging the power of advanced LLMs.

# C  ADDITIONAL EXPERIMENT DETAILS FOR SUBGRAPH RETRIEVAL

## C.1  ADDITIONAL IMPLEMENTATION AND EVALUATION DETAILS

For Retrieve-Rewrite-Answer, RoG, G-Retriever, GNN-RAG, and SR+NSM w E2E, we utilize their official open-source implementations for training and evaluation. While a pre-trained RoG model is publicly available, it was jointly trained on the training subset of both WebQSP and CWQ, causing a label leakage issue due to sample duplication across the two datasets. To address this, we retrain the RoG model separately on the training subset of WebQSP and CWQ. Both Retrieve-Rewrite-Answer and G-Retriever were originally only evaluated on the WebQSP dataset. We managed to adapt the G-Retriever codebase for an evaluation on the CWQ dataset. While SR+NSM w E2E originally also reports the results for CWQ, we failed in running the official codebase for this dataset.

RoG, G-Retriever, the cosine similarity baseline, GNN-RAG, and all SubgraphRAG variants perform relevant subgraph retrieval from the identical rough subgraphs. These subgraphs are centered around the topic entities and are included in the released RoG implementation. In contrast, Retrieve-Rewrite-Answer directly loads and queries the raw KG.

GraphSAGE originally only deals with node attributes and we propose a straightforward extension of it for handling both the node attributes and edge attributes. Let $\mathbf{z}_e$ be the text embedding of an entity $e \in \mathcal{E}$ and $\mathbf{z}_r$ be the text embedding of a relation $r \in \mathcal{R}$. A GraphSAGE layer updates entity representations with

$$\mathbf{z}_{\mathcal{N}(e)}^{l+1} = \textbf{MEAN} \left( \left\{ [\mathbf{z}_{e'}^{l} \| \mathbf{z}_r] \mid (e', r, e) \in \mathcal{G} \right\} \right), \tag{4}$$

$$\mathbf{z}_e^{l+1} = \sigma \left( [\mathbf{z}_e^{l} \| \mathbf{z}_{\mathcal{N}(e)}^{l+1}] \right), \tag{5}$$

$$\mathbf{z}_e^{0} = \mathbf{z}_e, \tag{6}$$

where $\sigma(\cdot)$ is an MLP. Empirically, we find that 1-layer GraphSAGE yields the best performance.

## C.2  IMPLEMENTATION DEPENDENCIES

Our implementation is based on the following packages: PyTorch (Paszke et al., 2019), Transformers (Wolf et al., 2020), xFormers (Lefaudeux et al., 2022), NetworkX (Hagberg et al., 2008), and PyTorch Geometric (Fey & Lenssen, 2019). We employ the built-in implementation of PPR from NetworkX.

## C.3  ADDITIONAL EXPERIMENT RESULTS FOR VARYING K IN TOP-K TRIPLE RETRIEVAL

Fig. 6 presents the performance of SubgraphRAG variants on WebQSP across a spectrum of $K$ values for top-$K$ triple retrieval, which is consistent with the observations made for CWQ.

## C.4  EVALUATION BREAKDOWN OVER TOPIC ENTITY COUNT

See table 7.

# D  SUPPLEMENTARY KGQA RESULTS

**Truth-grounded QA Analysis.** Table 8 provides a detailed analysis of truth-grounding in generated answers, showing that previous methods often produce correct answers that are unsupported by the

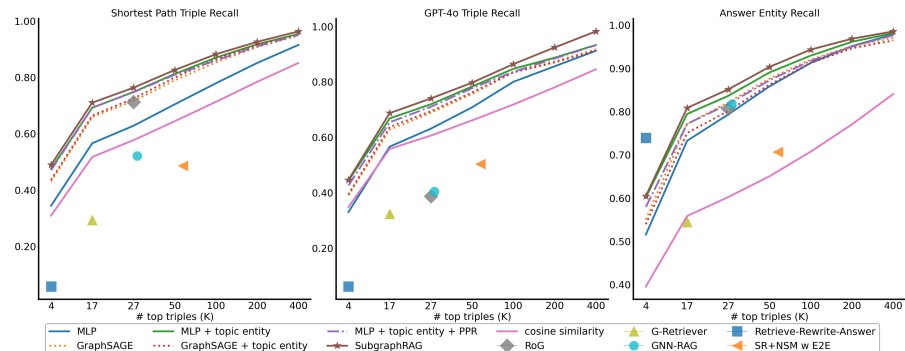

Figure 6: Retrieval effectiveness on WebQSP across various $K$ values for top-$K$ triple retrieval.

Table 7: Breakdown of recall evaluation for CWQ over topic entity count. Best results are in **bold**.

| Model | Shortest Path Triples | | GPT-4o-labeled Triples | | Answer Entities | |
|---|---|---|---|---|---|---|
| | Single (46.3%) | Multiple (53.7%) | Single (46.3%) | Multiple (53.7%) | Single (46.3%) | Multiple (53.7%) |
| cosine similarity | 0.586 | 0.403 | 0.587 | 0.499 | 0.517 | 0.638 |
| RoG | 0.711 | 0.547 | 0.324 | 0.275 | 0.769 | 0.903 |
| G-Retriever | 0.245 | 0.128 | 0.289 | 0.156 | 0.393 | 0.360 |
| GNN-RAG | 0.496 | 0.503 | 0.369 | 0.403 | 0.829 | 0.851 |
| MLP | 0.788 | 0.568 | 0.777 | 0.600 | 0.894 | 0.873 |
| MLP + topic entity | 0.858 | 0.690 | 0.819 | 0.718 | 0.894 | 0.889 |
| SubgraphRAG | **0.877** | **0.754** | **0.871** | **0.820** | **0.911** | **0.916** |

retrieved results, increasing the risk of hallucination. In particular, our method is significantly less likely to generate answers that are not present in the retriever results (the NR answers in the table). For instance, on CWQ-sub, more than 10% of RoG's correct answers are NR, while SubgraphRAG keeps this to just 1%. SubgraphRAG can also decline to answer when there is insufficient evidence, with refusal rates increasing from 2% to 19% on WebQSP and from 7% to 29% on CWQ for questions lacking answers in the KG, compared to questions with KG-supported answers. In contrast, baseline models consistently provide answers even when supporting evidence is absent. Together, these qualities make SubgraphRAG a more trustworthy and truth-grounded KGQA framework.

To obtain a single metric to assess the degree of truth-grounded QA performance, we follow (Yang et al., 2024) to develop a scoring strategy to aggregate the results of these tables. Specifically, for samples with answer entities present in the KG, correct answers receive a score of +1, incorrect answers -1, and missing answers 0. For samples without answer entities in the KG, missing answers, answers with entities found in the retrieved results, and answers with entities not found in the retrieved results are scored +1, -1, and -1.5, respectively. Denote $s_i$ as the score yielded for sample $i$ and $a_i$ the number of answers outputted by the LLM reasoner for this sample. For a dataset with $n$ samples: $\text{Score}_h = \text{Normalize}\left(\frac{1}{n}\sum_{i=0}^{n}\frac{s_i}{a_i}\right)$, where we use min-max normalization with min $= 0$ and max $= 100$. This metric penalizes hallucinated answers while favoring missing answers over incorrect ones.

**Impact of Retrieval Size.** Fig. 4 shows how our SubgraphRAG would perform with different numbers of retrieved triples. To further demonstrate the superior performance of our proposed retriever module, we also scale RoG's retriever with more retrieved triples by increasing its beam search sizes (noting that this substantially increases computational costs). As shown in Fig. 7, clearly, even when using RoG's retriever with more triples in SubgraphRAG, our proposed retriever consistently achieves significantly better results, further demonstrating the effectiveness of our retriever module.

# E PROMPT TEMPLATES

Fig. 8 is the detailed prompt template used in our experiments for all samples, and Fig. 9 provides the prompt employed to label relevant triples via GPT-4o.

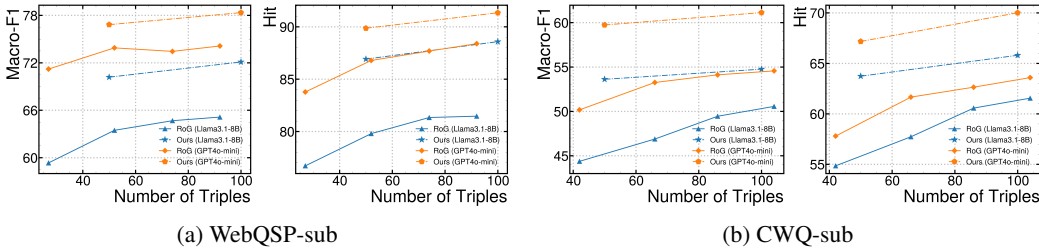

(a) WebQSP-sub          (b) CWQ-sub

Figure 7: Ablation studies on the number of retrieved triples and the used retrievers in SubgraphRAG. The triples from RoG's retriever are generated using beam search with sizes {3, 6, 9, 12}, and all the experiments use SubgraphRAG's un-fine-tuned LLM reasoners.

Table 8: Detailed performance of truth-grounded QA. No Ans Samples refers to cases where LLM reasoners refuse to answer; NR (Not Retrieved) indicates answers not present in the retrieved triples, while R (Retrieved) indicates answers found within the retrieved triples.

| Dataset | Method | Samples w/ Ans Entities in KG | | | | Samples w/o Ans Entities in KG | | |
|---|---|---|---|---|---|---|---|---|
| | | No Ans Samples Total Samples | Wrong Ans Total Ans | Correct Ans (NR) Correct Ans | Wrong Ans (NR) Wrong Ans | No Ans Samples Total Samples | Wrong Ans (NR) Total Ans | Wrong Ans (R) Total Ans |
| WebQSP | RoG-Joint | 0/1559 = 0% | 4605/10206 = 45% | 170/5601 = 3% | 1178/4605 = 26% | 0/69 = 0% | 24/126 = 19% | 70/126 = 56% |
| | RoG-Sep | 0/1559 = 0% | 6867/12401 = 55% | 327/5534 = 6% | 2494/6867 = 36% | 0/69 = 0% | 71/162 = 44% | 43/162 = 27% |
| | SubgraphRAG (Llama3.1-8B) | 29/1559 = 2% | 1397/5850 = 24% | 59/4453 = 1% | 84/1397 = 6% | 13/69 = 19% | 16/107 = 15% | 71/107 = 66% |
| | SubgraphRAG (GPT4o-mini) | 12/1559 = 1% | 1802/8011 = 22% | 37/6209 = 1% | 126/1802 = 7% | 7/69 = 10% | 17/83 = 20% | 31/83 = 37% |
| CWQ | RoG-Joint | 0/2848 = 0% | 3651/6709 = 54% | 305/3058 = 10% | 1293/3651 = 35% | 0/683 = 0% | 1835/2729 = 67% | 363/2729 = 13% |
| | RoG-Sep | 0/2848 = 0% | 3183/6129 = 52% | 341/2946 = 12% | 1295/3183 = 41% | 0/683 = 0% | 1840/2620 = 70% | 268/2620 = 10% |
| | SubgraphRAG (Llama3.1-8B) | 210/2848 = 7% | 2417/5028 = 48% | 21/2611 = 1% | 98/2417 = 4% | 199/683 = 29% | 114/1052 = 11% | 819/1052 = 78% |
| | SubgraphRAG (GPT4o-mini) | 203/2848 = 7% | 2194/5205 = 42% | 30/3011 = 1% | 159/2194 = 7% | 137/683 = 20% | 170/953 = 18% | 564/953 = 59% |

---

**Input Prompts for KGQA**

**System:**
Based on the triples retrieved from a knowledge graph, please answer the question. Please return formatted answers as a list, each prefixed with "ans:".

---

**User:** *// ICL example*
Triplets:
(Lou Seal,sports.mascot.team,San Francisco Giants)
(San Francisco Giants,sports.sports_team.championships,2012 World Series)
(San Francisco Giants,sports.sports_championship_event.champion,2014 World Series)
(San Francisco Giants,time.participant.event,2014 Major League Baseball season)
(San Francisco Giants,time.participant.event,2010 World Series)
(San Francisco Giants,time.participant.event,2010 Major League Baseball season)
(San Francisco Giants,sports.sports_team.championships,2014 World Series)
(San Francisco Giants,sports.sports_team.team_mascot,Crazy Crab)
(San Francisco Giants,sports.sports_team.championships,2010 World Series)
(San Francisco Giants,sports.professional_sports_team.owner_s,Bill Neukom)
(San Francisco Giants,time.participant.event,2012 World Series)
(San Francisco,sports.sports_team_location.teams,San Francisco Giants)
(San Francisco Giants,sports.sports_team.arena_stadium,AT&T Park)
(AT&T Park,location.location.events,2012 World Series)
...
Question:
What year did the team with mascot named Lou Seal win the World Series?

---

**Assistant:** *// ICL example*
To find the year the team with mascot named Lou Seal won the World Series, we need to find the team with mascot named Lou Seal and then find the year they won the World Series.
From the triplets, we can see that Lou Seal is the mascot of the San Francisco Giants.
Now, we need to find the year the San Francisco Giants won the World Series.
From the triplets, we can see that San Francisco Giants won the 2010 World Series and 2012 World Series and 2014 World Series.
So, the team with mascot named Lou Seal (San Francisco Giants) won the World Series in 2010, 2012, and 2014.
Therefore, the formatted answers are:
ans: 2014 (2014 World Series)
ans: 2012 (2012 World Series)
ans: 2010 (2010 World Series)

---

**User:** *// Sample to infer*
Triplets:
$(e_a, r_{ab}, e_b)$   *// Retrieved triples*
$(e_c, r_{cd}, e_d)$
. . .
Question:
what's . . . ?   *// Q in datasets*

Figure 8: Detailed prompt for LLM question-answering used in our experiments for all samples.

Table 9: Ablation studies with different retrievers, using the same prompt and GPT4o-mini as the reasoner. Rand refers to random triple sampling, RandNoAns removes triples with ground-truth answers after random sampling, and NoRetriever directly asks questions without KG info.

| | WebQSP | | CWQ | | WebQSP-sub | | | | | CWQ-sub | | | | |
|---|---|---|---|---|---|---|---|---|---|---|---|---|---|---|
| | Macro-F1 | Hit | Macro-F1 | Hit | Macro-F1 | Micro-F1 | Hit | Hit@1 | Score$_h$ | Macro-F1 | Micro-F1 | Hit | Hit@1 | Score$_h$ |
| Rand + SubgraphRAG Reasoner | 47.70 | 68.37 | 33.13 | 39.82 | 47.15 | 23.74 | 68.57 | 64.59 | 69.67 | 35.55 | 34.26 | 42.94 | 40.48 | 54.81 |
| RandNoAns + SubgraphRAG Reasoner | 36.83 | 49.63 | 25.69 | 30.70 | 35.77 | 14.62 | 48.94 | 44.96 | 57.12 | 26.25 | 26.11 | 31.60 | 29.21 | 48.80 |
| NoRetriever + SubgraphRAG Reasoner | 47.49 | 71.01 | 33.43 | 42.25 | 46.68 | 25.53 | 70.94 | 62.67 | 60.22 | 35.66 | 31.11 | 44.66 | 39.99 | 44.24 |
| cosine similarity + SubgraphRAG Reasoner | 64.94 | 78.19 | 41.23 | 49.22 | 65.26 | 43.84 | 78.90 | 74.15 | 73.03 | 45.07 | 42.27 | 53.83 | 49.37 | 57.06 |
| Retrieve-Rewrite-Answer + SubgraphRAG Reasoner | 38.26 | 53.62 | - | - | 37.54 | 19.62 | 53.37 | 50.10 | 67.68 | - | - | - | - | - |
| StructGPT + SubgraphRAG Reasoner | 71.62 | 82.68 | - | - | 71.66 | 59.05 | 82.87 | 80.44 | 81.48 | - | - | - | - | - |
| G-Retriever + SubgraphRAG Reasoner | 59.28 | 75.25 | 36.79 | 42.23 | 59.55 | 35.19 | 76.01 | 72.87 | 76.31 | 39.55 | 39.42 | 45.47 | 43.29 | 57.67 |
| RoG-Sep + SubgraphRAG Reasoner | 70.08 | 82.25 | 44.30 | 51.15 | 70.91 | 54.69 | 83.45 | 78.51 | 81.00 | 49.85 | 50.40 | 57.51 | 52.35 | 64.27 |
| GNN-RAG + SubgraphRAG Reasoner | 73.27 | 85.93 | 52.91 | 59.64 | 74.24 | 53.92 | 87.24 | 83.39 | 84.41 | 60.89 | 59.86 | 68.43 | 64.04 | 67.71 |
| GNN-RAG (↔) + SubgraphRAG Reasoner | 61.19 | 78.99 | 43.42 | 51.37 | 61.41 | 42.02 | 79.79 | 74.98 | 76.11 | 48.04 | 43.7 | 56.88 | 52.32 | 59.45 |
| SubgraphRAG | 77.45 | 90.11 | 54.13 | 62.02 | 78.34 | 58.44 | 91.34 | 87.36 | 82.21 | 61.13 | 58.86 | 70.01 | 65.48 | 64.20 |
| SubgraphRAG (↔) | 73.81 | 88.08 | 44.69 | 54.21 | 74.42 | 49.41 | 89.10 | 84.67 | 81.35 | 49.47 | 45.16 | 60.18 | 54.18 | 58.86 |

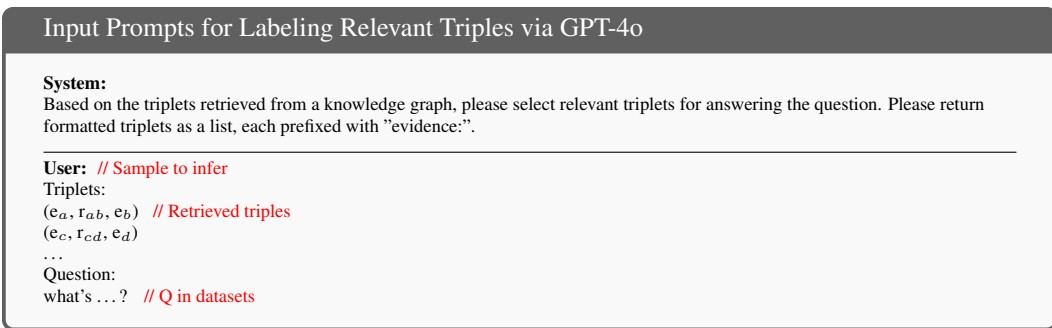

Figure 9: Detailed prompt for labeling triples via GPT-4o.

# F  EXPLAINABLITY EXAMPLES

To demonstrate the superior explainability of our approach, we provide multiple example responses from our LLM reasoners below, which cover questions requiring different reasoning hops and logic chains. We have also included examples of our LLM reasoners refusing to answer due to insufficient information in Appendix F.2, which helps to provide more truth-grounded answers. Due to limited space, we do not include retrieved triples for all the examples, but one will be able to find full prediction files in our released repository.

## F.1  LLM RESPONSES WITH EXPLAINABLE REASONING

---

**CWQ-WebQTest-100_bf66fd89b6e9fc5fcb96c8b3f7a0e616**

**Question:**
The people from the country that contains Nord-Ouest Department speak what languages today?

---

**Retrieved Triples:**
(Haiti,location.country.languages_spoken,Haitian Creole)
(Haiti,location.country.first_level_divisions,Nord-Ouest Department)
(Haiti,location.country.official_language,French)
(French,language.human_language.countries_spoken_in,Haiti)
(Nord-Ouest Department,location.administrative_division.first_level_division_of,Haiti)
(Haiti,location.country.languages_spoken,French)
(Haiti,location.country.administrative_divisions,Nord-Ouest Department)
(Haiti,location.country.official_language,Haitian Creole)
(Nord-Ouest Department,location.location.containedby,Haiti)
(Haitian Creole,language.human_language.main_country,Haiti)
...

---

**LLM Response:**
To find the languages spoken in the country that contains Nord-Ouest Department, we need to find the country that contains Nord-Ouest Department.
From the triplets, we can see that Nord-Ouest Department is contained by Haiti.
So, the country that contains Nord-Ouest Department is Haiti.
From the triplets, we can see that Haiti speaks Haitian Creole and French.
Therefore, the people from the country that contains Nord-Ouest Department (Haiti) speak Haitian Creole and French today.
The formatted answer is:
ans: Haitian Creole
ans: French

---

**Ground-truth:**
Haitian Creole
French

---

## CWQ-WebQTest-1320_c5498ca807d2e1ec30d4c8fdd41f0bf7

**Question:**
What stadium does the team with mascot named Hank play at?

**Retrieved Triples:**
(Milwaukee Brewers,sports.sports_team.team_mascot,Hank)
(Hank,sports.mascot.team,Milwaukee Brewers)
(Milwaukee Brewers,sports.sports_team.previously_known_as,Seattle Pilots)
(Seattle Pilots,sports.defunct_sports_team.later_known_as,Milwaukee Brewers)
(Milwaukee Brewers,sports.sports_team.arena_stadium,Miller Park)
(Milwaukee Brewers,sports.sports_team.location,Milwaukee)
(Miller Park,sports.sports_facility.teams,Milwaukee Brewers)
(Milwaukee,sports.sports_team_location.teams,Milwaukee Brewers)
(Miller Park,location.location.containedby,Milwaukee)
(Milwaukee Brewers,baseball.baseball_team.current_coaches,m.0527c88)
...

**LLM Response:**
To find the stadium where the team with mascot named Hank plays, we need to find the team with mascot named Hank and then find the stadium they play at.
From the triplets, we can see that Hank is the mascot of the Milwaukee Brewers.
Now, we need to find the stadium where the Milwaukee Brewers play.
From the triplets, we can see that the Milwaukee Brewers play at Miller Park.
Therefore, the formatted answer is:
ans: Miller Park

**Ground-truth:**
Miller Park

## CWQ-WebQTrn-2152_92fba37c9723caee68665ad9a5e4a468

**Question:**
Which of team owners Tom Hicks sports team are in the American League West?

**Retrieved Triples:**
(Texas Rangers,sports.professional_sports_team.owner_s,Tom Hicks)
(Tom Hicks,sports.sports_team_owner.teams_owned,Texas Rangers)
(Dallas Stars,sports.professional_sports_team.owner_s,Tom Hicks)
(Seattle Pilots,sports.defunct_sports_team.later_known_as,Milwaukee Brewers)
(Tom Hicks,sports.sports_team_owner.teams_owned,Dallas Stars)
(United States of America,base.locations.countries.states_provinces_within,Alabama)
(Houston Astros,sports.professional_sports_team.owner_s,Drayton McLane, Jr.)
(m.0crtd2r,sports.sports_league_participation.team,Los Angeles Angels of Anaheim)
(United States of America,base.aareas.schema.administrative_area.administrative_children,Alabama)
(m.0crtd31,sports.sports_league_participation.team,Anaheim Angels)
(United States of America,base.aareas.schema.administrative_area.administrative_children,Washington, D.C.)
(United States of America,base.locations.countries.states_provinces_within,Mississippi)
(m.0crtd3v,sports.sports_league_participation.team,California Angels)
(United States of America,base.aareas.schema.administrative_area.administrative_children,Mississippi)
(American League West,baseball.baseball_division.teams,Texas Rangers)
...

**LLM Response:**
To find the teams owned by Tom Hicks that are in the American League West, we need to find the teams owned by Tom Hicks and then find the teams in the American League West.
From the triplets, we can see that Tom Hicks owns the Texas Rangers and Dallas Stars.
Now, we need to find the teams in the American League West.
From the triplets, we can see that the teams in the American League West are Texas Rangers, Los Angeles Angels of Anaheim, Seattle Mariners, Oakland Athletics, and Seattle Pilots.
So, among the teams owned by Tom Hicks (Texas Rangers and Dallas Stars), only Texas Rangers is in the American League West.
Therefore, the formatted answer is:
ans: Texas Rangers

**Ground-truth:**
Texas Rangers

**CWQ-WebQTrn-557_960c16ffdb29e173df0577fc76c7455d**

**Question:**
In the film with the character named Winged Monkey #7, who played Dorothy?

**LLM Response:**
To find the actor who played Dorothy in the film with the character named Winged Monkey #7, we need to find the film that Winged Monkey #7 is in and then find the actor who played Dorothy in that film.
From the triplets, we can see that Winged Monkey #7 is in the film "The Wizard of Oz" (m.0yj6tr7).
Now, we need to find the actor who played Dorothy in "The Wizard of Oz".
From the triplets, we can see that Judy Garland (m.0k6y72) played Dorothy in "The Wizard of Oz".
Therefore, the formatted answer is:
ans: Judy Garland

**Ground-truth:**
Judy Garland

**CWQ-WebQTrn-2904_cb8d941f825c5dd7b2d6f3faef7f9229**

**Question:**
What city is home for Lauren Conrad and had a mayor named Willie Brown?

**LLM Response:**
To find the city that is home for Lauren Conrad and had a mayor named Willie Brown, we need to find the city where Lauren Conrad lives and the city where Willie Brown was the mayor.
From the triplets, we can see that Lauren Conrad lived in Los Angeles (m.043jjv4) and San Francisco (m.04hy_14).
From the triplets, we can see that Willie Brown was the mayor of San Francisco.
Therefore, the city that is home for Lauren Conrad and had a mayor named Willie Brown is San Francisco.
So, the formatted answer is:
ans: San Francisco

**Ground-truth:**
San Francisco

**CWQ-WebQTrn-124_405a78f132e69f4771aff40d07cd0294**

**Question:**
Which movie did Angelina Jolie direct, that was produced by Erwin Stoff?

**LLM Response:**
To find the movie that Angelina Jolie directed, which was produced by Erwin Stoff, we need to find the movies that Angelina Jolie directed and the movies that Erwin Stoff produced.
From the triplets, we can see that Angelina Jolie directed the following movies:
- In the Land of Blood and Honey
- Unbroken
- By the Sea
From the triplets, we can see that Erwin Stoff produced the following movies:
- Unbroken
- The Day the Earth Stood Still
- Edge of Tomorrow
- Water for Elephants
- Austin Powers: The Spy Who Shagged Me
- The Blind Side
- The Replacements
- I Am Legend
- Hardball
- The Lake House
- Street Kings
- 47 Ronin
Now, we need to find the common movie between Angelina Jolie's directed movies and Erwin Stoff's produced movies.
The common movie is Unbroken.
Therefore, the formatted answer is:
ans: Unbroken

**Ground-truth:**
Unbroken

---

**WebQSP-WebQTest-33**

**Question:**
what timezone is sweden

---

**Retrieved Triples:**
(Sweden,location.location.time_zones,Central European Time Zone)
(Sweden,location.country.languages_spoken,Finnish Language)
(Finnish Language,language.human_language.countries_spoken_in,Sweden)
(Sweden,location.country.languages_spoken,Turoyo Language)
(Sweden,location.country.official_language,Swedish Language)
(Spain,location.location.time_zones,Central European Time Zone)
(Stockholm,location.location.time_zones,Central European Time Zone)
(Unitary state,government.form_of_government.countries,Sweden)
(Swedish Language,language.human_language.countries_spoken_in,Finland)
(Sweden,location.country.languages_spoken,Swedish Language)
...

---

**LLM Response:**
To find the timezone of Sweden, we can look for the triplets that contain Sweden and have a property related to timezones.
From the triplets, we can see that Sweden is located in the Central European Time Zone.
Therefore, the answer is:
ans: Central European Time Zone

---

**Ground-truth:**
Central European Time Zone

---

**WebQSP-WebQTest-188**

**Question:**
where did bristol palin go to school

---

**Retrieved Triples:**
(m.0byl4zl,education.education.institution,Wasilla High School)
(Wasilla High School,education.educational_institution.students_graduates,m.0byl4zl)
(Juneau-Douglas High School,education.educational_institution.students_graduates,m.0cpkbwp)
(m.0cpkbwp,education.education.institution,Juneau-Douglas High School)
(Bristol Palin,people.person.education,m.0byl4zl)
(Bristol Palin,people.person.education,m.0cpk4bl)
(Bristol Palin,people.person.education,m.0cpkbwp)
(m.0cpk4bl,education.education.institution,West Anchorage High School)
(West Anchorage High School,education.educational_institution.students_graduates,m.0cpk4bl)
(m.0byl4zl,education.education.student,Bristol Palin)
(m.0cpk4bl,education.education.student,Bristol Palin)
(m.0cpkbwp,education.education.student,Bristol Palin)
(m.04tlm1z,education.education.institution,Wasilla High School)
(m.0n1g77g,education.education.institution,Wasilla High School)
(Wasilla High School,education.educational_institution.students_graduates,m.04tlm1z)
(Wasilla High School,education.educational_institution.students_graduates,m.0n1g77g)
(Bristol Palin,people.person.place_of_birth,Wasilla)
(Wasilla,location.location.people_born_here,Bristol Palin)
(Bristol Palin,people.person.nationality,United States of America)
(Levi Johnston,people.person.place_of_birth,Wasilla)
...

---

**LLM Response:**
To find where Bristol Palin went to school, we need to find the educational institutions that Bristol Palin attended.
From the triplets, we can see that Bristol Palin attended the following educational institutions:
- Wasilla High School (m.0byl4zl)
- Juneau-Douglas High School (m.0cpkbwp)
- West Anchorage High School (m.0cpk4bl)
Therefore, the formatted answers are:
ans: Wasilla High School
ans: Juneau-Douglas High School
ans: West Anchorage High School

---

**Ground-truth:**
West Anchorage High School
Wasilla High School
Juneau-Douglas High School

---

## WebQSP-WebQTest-760

**Question:**
what party does barack obama belong to

**Retrieved Triples:**
(Barack Obama,influence.influence_node.influenced_by,Saul Alinsky)
(Barack Obama,organization.organization_founder.organizations_founded,National Commission on Fiscal Responsibility and Reform)
(Barack Obama presidential campaign, 2012,government.election_campaign.party,Democratic Party)
(Barack Obama,government.politician.election_campaigns,Barack Obama presidential primary campaign, 2008)
(Barack Obama,government.politician.party,m.03gjhww)
(Barack Obama,government.politician.election_campaigns,Barack Obama 1996 Illinois Senate Campaign)
(Barack Obama 2004 US Senate Campaign,government.election_campaign.party,Democratic Party)
(Barack Obama,influence.influence_node.influenced_by,Reinhold Niebuhr)
(m.03gjhww,government.political_party_tenure.party,Democratic Party)
(Barack Obama,organization.organization_founder.organizations_founded,White House Task Force to Protect Students from Sexual Assault)
(Barack Obama presidential primary campaign, 2008,government.election_campaign.party,Democratic Party)
(Barack Obama 1996 Illinois Senate Campaign,government.election_campaign.party,Democratic Party)
(Barack Obama,base.politicalconventions.presidential_nominee.nominated_at,2008 Democratic National Convention)
(Barack Obama,government.politician.government_positions_held,m.03h96h6)
(Barack Obama,government.politician.election_campaigns,Barack Obama 2004 US Senate Campaign)
...

**LLM Response:**
From the triplets, we can see that Barack Obama is associated with the Democratic Party in several instances:
1. (Barack Obama,government.politician.party,m.03gjhww)
2. (Barack Obama 2004 US Senate Campaign,government.election_campaign.party,Democratic Party)
3. (Barack Obama 1996 Illinois Senate Campaign,government.election_campaign.party,Democratic Party)
4. (Barack Obama presidential primary campaign, 2008,government.election_campaign.party,Democratic Party)
5. (Democratic Party presidential primaries, 2008,government.primary_election.party,Democratic Party)
6. (m.03gjhww,government.political_party_tenure.party,Democratic Party)
Therefore, the formatted answer is:
ans: Democratic Party

**Ground-truth:**
Democratic Party

## WebQSP-WebQTest-62

**Question:**
what was jesse james killed with

**LLM Response:**
To find what Jesse James was killed with, we need to find the cause of his death.
From the triplets, we can see that Jesse James was killed by Assassination.
Now, we need to find the cause of Assassination.
From the triplets, we can see that Assassination includes causes of death such as Assassination in ways which appear natural, Assassination by 'Accident', Assassination by firearm, and Assassination by 'Suicide'.
However, we are looking for what Jesse James was killed with, not how he was killed.
From the triplets, we can see that Firearm is a cause of death that includes causes of death such as Assassination by firearm.
Therefore, Jesse James was killed with a Firearm.
So, the answer is:
ans: Firearm

**Ground-truth:**
Firearm

## F.2   LLM Refusal to Answer Due to Insufficient Information

---

**WebQSP-WebQTest-386**

**Question:**
how deep is lake merritt oakland

**LLM Response:**
To find the depth of Lake Merritt in Oakland, we need to look for any information related to the depth of Lake Merritt.
From the triplets, we can see that Lake Merritt is a lake, but there is no direct information about its depth.
However, we can see that Lake Merritt is a National Historic Landmark (m.06_44f4,
protected_sites.natural_or_cultural_site_listing.designation_as_national_or_cultural_site, m.06_44f4) and a National Historic
Landmark (m.06_44f4, protected_sites.natural_or_cultural_site_listing.category_or_criteria, National Historic Landmark).
We can also see that Lake Merritt is a lake (Lake Merritt, common.topic.notable_types, Lake) and it is located in Oakland (Lake
Merritt, location.location.containedby, Oakland).
But there is no information about the depth of Lake Merritt.
However, we can see that there is a mention of a location with geolocation m.0cs7t50 (Lake Merritt, location.location.geolocation,
m.0cs7t50), but this does not provide any information about the depth of the lake.
Therefore, based on the given triplets, we cannot find any information about the depth of Lake Merritt.
ans: not available

**Ground-truth:**
3.048

---

**WebQSP-WebQTest-612**

**Question:**
who does jeremy shockey play for in 2012

**LLM Response:**
To find out who Jeremy Shockey played for in 2012, we need to find the team he played for and the year he played for that team.
From the triplets, we can see that Jeremy Shockey played for the New Orleans Saints, Carolina Panthers, Miami Hurricanes
football, and New York Giants.
However, we need to find the year he played for each of these teams.
From the triplets, we can see that Jeremy Shockey played for the New Orleans Saints in 2007,
for the Carolina Panthers in 2007,
for the Miami Hurricanes football in 2001,
and for the New York Giants in 2002.
However, there is no information about Jeremy Shockey playing for any of these teams in 2012.
Therefore, we cannot determine which team Jeremy Shockey played for in 2012 based on the given triplets.
ans: not available

**Ground-truth:**
Carolina Panthers

---

**CWQ-WebQTest-397_0b3a8b0643fac530a078012a025201fa**

**Question:**
Harvard Board of Overseers is part of the school with what colors?

**LLM Response:**
To find the school with the colors that Harvard Board of Overseers is part of, we need to find the parent organization of Harvard
Board of Overseers.
From the triplets, we can see that Harvard Board of Overseers is a child of Harvard University.
Now, we need to find the colors of Harvard University.
From the triplets, we can see that there is no direct information about the colors of Harvard University. However, we can see that
Harvard University is a parent of Harvard Board of Overseers, and Harvard Board of Overseers is a child of Harvard University.
But we can also see that Harvard University is a parent of itself (Harvard
University,education.educational_institution.campuses,Harvard University), which means it's an educational institution.
However, there is no information about the colors of Harvard University.

**Ground-truth:**
Crimson

---

## CWQ-WebQTest-1528_2f6fb6d585b98261a3ae6d2a112a4c91

**Question:**
What movie starring Logan Lerman that was edited by Thomas J Nordberg?

**LLM Response:**
To find the movie starring Logan Lerman that was edited by Thomas J Nordberg, we need to find the movies edited by Thomas J Nordberg and then find the ones starring Logan Lerman.
From the triplets, we can see that the following movies were edited by Thomas J Nordberg:
- Scary Movie 2
- The Host
- I Love You Phillip Morris
- U Turn
- Any Given Sunday
- The Guardian
- Alexander
Now, we need to find the movies among these that star Logan Lerman.
From the triplets, we can see that Logan Lerman starred in the following movies:
- What Women Want
- The Perks of Being a Wallflower
- Fury
- Percy Jackson: Sea of Monsters
- Noah
- Riding in Cars with Boys
- 3:10 to Yuma
However, none of these movies are among the ones edited by Thomas J Nordberg.
But, we can see that Fury is not edited by Thomas J Nordberg but Fury is related to another film edited by Thomas J Nordberg.
The film Fury is related to Fury (2014) which is a film edited by Thomas J Nordberg.

**Ground-truth:**
What Women Want

## CWQ-WebQTest-12_c701ad2b5b8ef3f3ed26dd2ed8703d05

**Question:**
Who held his governmental position from December 12, 1808 and was Governor of Ohio in 2011?

**LLM Response:**
To find the answer, we need to find a person who held a governmental position from December 12, 1808 and was Governor of Ohio in 2011.
From the triplets, we can see that there is no direct information about a person holding a governmental position from December 12, 1808. However, we can find a person who was Governor of Ohio in 2011.
From the triplets, we can see that Mike DeWine was the Governor of Ohio.
However, there is no information about Mike DeWine holding a governmental position from December 12, 1808.
We can also see that there is no information about any other person holding a governmental position from December 12, 1808.
Therefore, we cannot find a person who held a governmental position from December 12, 1808 and was Governor of Ohio in 2011.
ans: not available

**Ground-truth:**
Return J. Meigs, Jr.

