# OpenReview forum: "Simple is Effective: The Roles of Graphs and Large Language Models in Knowledge-Graph-Based Retrieval-Augmented Generation"
_ICLR.cc/2025/Conference — ICLR 2025 Poster_

### Official Review · Reviewer_q2Wy · 2024-10-21

**Soundness:** 4
**Presentation:** 4
**Contribution:** 3
**Rating:** 8
**Confidence:** 5

**Summary:**

This paper proposes a novel KBQA method named SubgraphRAG. Instead of incorporating more powerful LLMs and/or advanced SFT techniques, SubgraphRAG focuses on the retrieval part. It extends the existing KG-based RAG framework with an MLP and a newly proposed triple-scoring mechanism, which shows to be effective and efficient. The retrieved subgraphs allow the proposed method to achieve SOTA performance on smaller LLMs without SFT.

**Strengths:**

1) The problem is clearly defined. (Section 3.1, equation 1 and 2)
2) The proposed method is easy to follow. It also includes several details for the reader to
reproduce the results, e.g. the initialization of embeddings, the introduction of DDE.
3) The proposed retrieval method is efficient and effective. Better retrievers enhance the
performance ceiling of RAG.
4) This paper includes detailed ablation study (Q1, Q2, and Q5), which proves the effective-
ness of the retriever part.
5) The proposed method achieves SOTA performance on low-scale pretrained LLMs.

**Weaknesses:**

1) This reviewer suggests the authors move the main results and related analysis to an earlier
subsection of the experiments part, and group the ablation studies together.
2) The WebQSP dataset does not include official explanations for its answers. However, the proposed method requires explanations to perform reliable in-context learning, which relies on external knowledge other than the dataset or the KG.

**Questions:**

1) As mentioned in Section 3.2, the authors mentioned that the designed prompt template is shown in Figure 2. In addition, this reviewer is interested to know some more *details* about the design of “in-context demonstrations”.
a) Does SubgraphRAG utilize the same “in-context demonstration” for all the questions?
b) How were these in-context demonstrations designed? Specifically, what are the criteria for the selection of these demonstrations? How do you generate the explanations?

2) The authors raise concerns about the reproducibility issue of the ToG baseline. Since ToG relies on closed-source LLMs such as GPT4, this reviewer believes that the performance can be greatly affected by the specific versions of LLMs. SubgraphRAG also uses closed-source LLMs for evaluation. Which specific version of GPT4o is being used in this work? (Please also include this in the *implementation* part in the paper.)

---

> ### Author Response · Authors · 2024-11-22
>
> Thank you for your insightful and positive feedback! Below we respond to your questions and concerns.
>
> **Q1**: As mentioned in Section 3.2, the authors mentioned that the designed prompt template is shown in Figure 2. In addition, this reviewer is interested to know some more details about the design of “in-context demonstrations”. a) Does SubgraphRAG utilize the same “in-context demonstration” for all the questions? b) How were these in-context demonstrations designed? Specifically, what are the criteria for the selection of these demonstrations? How do you generate the explanations?
>
> **Response**:
> a) Yes, all questions use the same in-context demonstration.
>
> b) The in-context demonstration is designed with the goal of encouraging LLMs to produce reasoning that is grounded in evidence. To achieve this, we crafted a template that includes a reasoning chain as part of the demonstration. We want this template to serve to guide the LLMs in generating explainable QA results by illustrating how reasoning with evidence can lead to the correct answer. For a concrete example, you may refer to Figure 7 in Appendix.
>
> **Q2**: The authors raise concerns about the reproducibility issue of the ToG baseline. Since ToG relies on closed-source LLMs such as GPT4, this reviewer believes that the performance can be greatly affected by the specific versions of LLMs. SubgraphRAG also uses closed-source LLMs for evaluation. Which specific version of GPT4o is being used in this work? (Please also include this in the implementation part in the paper.)
>
> **Response**: We thank the reviewer for raising this point. We used gpt-3.5-turbo-1106, gpt-4o-mini-2024-07-18, and gpt-4o-2024-08-06 in our experiments. We have added this statement in Sec. 4.2 in the revised manuscript.

---

> > ### Comment · Reviewer_q2Wy · 2024-11-23
> > **Reply to the rebuttal.**
> >
> > Thank you for your response.
> >
> > Regarding the selection of in-context demonstrations, this reviewer believes that using examples tailored to the problem can further improve performance compared to using the same set of demonstrations for all cases.
> >
> > Please ensure the mentioned parts will be included in the camera-ready version.
> > This reviewer has no additional concerns and has decided to maintain the ratings.

---

> > > ### Author Response · Authors · 2024-11-23
> > >
> > > Thank you for reading our response! We're glad that we've addressed your concerns, and we'll prepare the manuscript as you suggested.

---

### Official Review · Reviewer_y2dZ · 2024-10-27

**Soundness:** 2
**Presentation:** 3
**Contribution:** 2
**Rating:** 5
**Confidence:** 5

**Summary:**

The paper presents SubgraphRAG, a graph retrieval method for Knowledge Graph Question Answering (KGQA). SubgraphRAG trains a retrieval module to score relevant triplets of the KG based on training questions. During inference, SubgraphRAG retrieves top-scored relevant triplets based on the question and uses them as additional context to the LLM for KGQA. Experiments are performs on WebQSP and CWQ benchmarks, evaluating retrieval and downstream KGQA performance.

**Strengths:**

The paper strengths are summarized below:

- S1) The retrieval module is lightweight pre-trained text encoder, followed by MLPs. This allows fast retrieval and does not incur significant latency during KGQA with LLMs.
- S2) The experimentation includes ablation studies in both retrieval and downstream KGQA performance, which are helpful to understand the benefits of each of the competing methods.
- S3) The paper reads nicely and is easy to follow.

**Weaknesses:**

The main weaknesses of the paper are summarized below:

- W1) SubgraphRAG emphasizes the necessity for  lightweight retrieval for KGQA. However, GNNs are already established as lightweight retrievers (as also shown in Table 1) and specifically designed for handling KGs. Although the paper references some GNN-based approaches [1,2], it does not compare SubgraphRAG with them. Without these comparisons, it is unclear why SubgraphRAG favors MLPs over GNNs and what the rationale behind this choice is.

- W2) The retrieval evaluation, as presented in Table 1, relies solely on the recall metric. However, SubgraphRAG retrieves a larger number of triplets (e.g., top-100), which should naturally achieve higher recall than the baselines. To ensure a fair comparison, additional metrics such as precision, F1 score, or recall @ k should also be included. Furthermore, in the KGQA results, SubgraphRAG utilizes more advanced LLMs, such as GPT-4o, compared to the baselines. This raises questions about the advantages of SubgraphRAG when using the same LLMs. For instance, SubgraphRAG combined with LLaMa 3.1-8B does not outperform RoG in the CWQ dataset.

[1] EtD: Explore then determine: A gnn-llm synergy framework for reasoning over knowledge graph. arXiv preprint arXiv:2406.01145, 2024a.
[2] GNN-RAG: Graph neural retrieval for large language model reasoning. arXiv preprint arXiv:2405.20139, 2024.

**Questions:**

Please refer to the previous comments. Additionally, I have few further questions/comments:
- Q1) Could you provide some citations on what "locality-sensitive hashing, designed for similarity search" in Line 076 means?
- Q2) Line 101 (necessity for lightweight retrieval) and the final choice of MLP contradict Line 095 (LSTMs and GNNs are limited because their lightweight).
- Q3) In Line 323, SubgraphRAG  continues to use shortest paths, despite their limitations noted in Line 097.
- Q4) The prompt for GPT-4o scoring (Fig.7) does not include the ground-truth answer. Could the evaluation results change if we include the ground-truth answers?
- Q5) Line 429 mentions that RoG suffers from label leakage. However, SubgraphRAG utilized GPT-4o, whose training data is unknown and could also be affected by training leakage.

---

> ### Author Response · Authors · 2024-11-22
>
> Thank you for the valuable constructive feedback! Below we respond to your questions and concerns.
>
> **Q1**: The retrieval evaluation, as presented in Table 1, relies solely on the recall metric. However, SubgraphRAG retrieves a larger number of triplets (e.g., top-100), which should naturally achieve higher recall than the baselines. To ensure a fair comparison, additional metrics such as precision, F1 score, or recall @ k should also be included.
>
> **Response**: We appreciate this important observation about evaluation metrics. In Figures 3 and 5, we plot retrieval recall values against varying numbers of retrieved triples (recall @ k). These results demonstrate that SubgraphRAG **consistently outperforms baseline retrievers when operating under identical retrieval budgets**. Since we maintain controlled retrieval sizes, superior recall directly implies better precision and F1. Furthermore, our analysis reveals that SubgraphRAG exhibits favorable **scaling properties**: as more top-ranked triples are incorporated, the recall improvements remain substantial. This scaling characteristic is particularly valuable for accommodating the varying long-context reasoning capabilities of different LLMs.
>
> **Q2**: Is SubgraphRAG still superior if we control the LLM used for reasoning? In particular, it does not outperform RoG with Llama 2-7B on CWQ when combined with Llama 3.1-8B.
>
> **Response**: We appreciate the reviewer’s astute observation. While the observation made is correct, it’s crucial to understand the root cause.
>
> To rigorously address this concern, we conducted comprehensive ablation studies that isolate the retriever’s impact by controlling the LLM reasoner. Our experiments employ Llama 3.1-8B and more powerful GPT-4o-mini as **fixed LLM reasoners while varying only the retrieval method**. The results below demonstrate that **SubgraphRAG consistently outperforms RoG** across both LLM reasoners and datasets.
>
> **Llama 3.1-8B + WebQSP**
>
> | Retriever  | F1 | Hit |
> |--|----|---|
> | RoG | 63.54 | 79.42 |
> | SubgraphRAG | **70.57** | **86.61** |
>
> **Llama 3.1-8B + CWQ**
>
> | Retriever  | F1 | Hit |
> |--|----|---|
> | RoG | 42.66 | 52.00 |
> | SubgraphRAG | **47.16** | **56.98** |
>
> **GPT-4o-mini + WebQSP**
>
> | Retriever  | F1 | Hit |
> |--|----|---|
> | RoG | 73.21 | 87.04 |
> | SubgraphRAG | **77.45** | **90.11** |
>
> **GPT-4o-mini + CWQ**
>
> | Retriever  | F1 | Hit |
> |--|----|---|
> | RoG | 48.02 | 55.88 |
> | SubgraphRAG | **54.13** | **62.02** |
>
> Regarding the observation made in the question for comparing against RoG + a 7B LLM, it’s important to note that the 7B LLM employed by RoG has been **directly fine-tuned** on CWQ. In contrast, our approach only **prompts pre-trained** LLMs, unlocking the potential in adopting advanced closed-source LLMs while maintaining generalizability. The controlled experiments provide strong evidence that SubgraphRAG's performance improvements stem substantially from its enhanced retrieval capabilities.
>
> **Q3**: How well does SubgraphRAG perform when compared to alternative lightweight GNN-based approaches like EtD and GNN-RAG?
>
> **Response**: We’ve performed additional empirical studies and updated the manuscript accordingly. The studies demonstrate the superior performance when compared to GNN-based approaches.
>
> For **retrieval evaluation**, we are only able to compare against GNN-RAG as the official implementation of EtD remains unavailable. GNN-RAG employs two GNNs to predict answer entities and then extracts shortest paths between topic entities and predicted answer entities. Our retriever consistently outperforms GNN-RAG (Table 1, Figure 3, and Figure 5). Notably, **while our retriever is not explicitly trained to predict answer entities, it outperforms GNN-RAG in answer entity recall using equivalent retrieval budgets** (Figure 3 and 5). In terms of efficiency, our retriever is **an order of magnitude faster** than GNN-RAG (Table 1).
>
> For **end-to-end KGQA evaluation**, SubgraphRAG consistently outperforms EtD and performs better or comparable with respect to GNN-RAG (Table 3). As GNN-RAG adopts the strategy of RoG in fine-tuning LLM reasoners, we again perform the studies controlling the LLM reasoner as in the response to Q2. In this more controlled setting, our approach consistently outperforms GNN-RAG.
>
> **Llama 3.1-8B + WebQSP**
>
> | Retriever  | F1 | Hit |
> |--|----|---|
> | GNN-RAG | 62.34 | 78.81 |
> | SubgraphRAG | **70.57** | **86.61** |
>
> **Llama 3.1-8B + CWQ**
>
> | Retriever  | F1 | Hit |
> |--|----|---|
> | GNN-RAG | 46.40 | 54.35 |
> | SubgraphRAG | **47.16** | **56.98** |
>
> **GPT-4o-mini + WebQSP**
>
> | Retriever  | F1 | Hit |
> |--|----|---|
> | GNN-RAG | 73.27 | 85.93 |
> | SubgraphRAG | **77.45** | **90.11** |
>
> **GPT-4o-mini + CWQ**
>
> | Retriever  | F1 | Hit |
> |--|----|---|
> | GNN-RAG | 52.91 | 59.64 |
> | SubgraphRAG | **54.13** | **62.02** |

---

> > ### Author Response · Authors · 2024-11-22
> >
> > Overall, the empirical studies demonstrate the surprising **effectiveness and efficiency** when compared to dedicated GNN-based approaches. The performance gains may be attributed to fundamental limitations of GNNs, including limited representation power for structural information [1] and degradation of pre-trained text embeddings due to over-smoothing [2] and over-squashing [3].
> >
> > **Q4**: Why does SubgraphRAG continue to employ shortest paths for supervision signals, despite their limitations noted?
> >
> > **Response**: High-quality supervision signals require manual labeling and are often not readily available. Consequently, previous works like RoG adopt shortest paths for weak supervision signals [4]. We admit that shortest paths may not be the best supervision signals. To demonstrate that the better performance of our framework indeed comes from the framework design by itself rather than potentially high-quality labels, we still use shortest paths.
> >
> > Despite the usage of shortest paths, SubgraphRAG is not constrained by this choice as the retrieved subgraphs could have flexible sizes and do not necessarily follow a path type. This can be demonstrated by the significantly better performance of SubgraphRAG in retrieving triples identified as relevant by GPT-4o, when compared to RoG. This suggests that our approach effectively transcends the limitations of its training signals. See Section 4.1 for the details.
> >
> > **Q5**: While the authors state that RoG suffers from label leakage, SubgraphRAG utilized GPT-4o, whose training data is unknown and could also be affected by training leakage.
> >
> > **Response**: As demonstrated in our response to Q2, **SubgraphRAG maintains superior performance even when both approaches utilize GPT-4o-mini**, suggesting that our improvements stem substantially from retriever advantages. While we acknowledge that LLM training data opacity affects all KG-based RAG studies, the specific concern about RoG's label leakage arises from their joint LLM fine-tuning over WebQSP and CWQ training subsets. This is why we introduced RoG-Sep, which trains separate models for each dataset. We are grateful to the RoG authors for their well-documented codebase, which enabled this additional analysis.
> >
> > **Q6**: The prompt for GPT-4o scoring (Fig.7) does not include the ground-truth answer. Could the evaluation results change if we include the ground-truth answers?
> >
> > **Response**: Thank you for raising this point. Including the ground-truth answers in the prompt could indeed influence the results. The reason we chose not to include ground-truth answers in the prompt was to avoid potential statistical bias. Specifically, we think the presence of answer tokens in the prompt might cause the LLMs to statistically favor triples containing the answer entities, even if some of those triples could be less relevant. Our goal is to encourage the LLMs to prioritize reasoning chains and select triples based on their relevance rather than the presence of specific answer entities. Nonetheless, we agree that further investigation into the scoring strategies with LLMs would be valuable.
> >
> > **Q7**: The introduction argues that existing lightweight retrievers based on LSTMs or GNNs are insufficient due to the constrained reasoning capacity of lighter models, which seems self-contradictory with the argued design consideration for lightweight retrieval.
> >
> > **Response**: We appreciate this observation regarding the seeming contradiction. Several previous works employ GNNs or LSTMs for both retrieval and reasoning. Our argument rests on the distinction between retrieval and reasoning - while lightweight models can excel at retrieval, they often struggle when tasked with complex reasoning. We’ve revised the introduction to enhance clarity.
> >
> > **Q8**: Could you provide some citations on what "locality-sensitive hashing, designed for similarity search" means?
> >
> > **Response**: At a high level, locality-sensitive hashing is an approach for efficient approximate nearest neighbor search, where similar candidates get hashed into the same buckets with high probability. We’ve updated the manuscript to include a citation for a classical paper [5].
> >
> > [1] Li et al. Distance Encoding: Design Provably More Powerful Neural Networks for Graph Representation Learning.
> >
> > [2] Oono & Suzuki. Graph Neural Networks Exponentially Lose Expressive Power for Node Classification.
> >
> > [3] Topping et al. Understanding over-squashing and bottlenecks on graphs via curvature.
> >
> > [4] Luo et al. Reasoning on Graphs: Faithful and Interpretable Large Language Model Reasoning.
> >
> > [5] Indyk & Motwani. Approximate nearest neighbors: towards removing the curse of dimensionality.

---

> ### Comment · Reviewer_y2dZ · 2024-11-23
>
> Thank you to the authors for their detailed response and for the effort put into addressing the reviewers' comments. I also noticed that the manuscript has been updated (e.g., Figure 3 appears to be new). To help reviewers track changes, I would recommend that the authors document the modifications between the two versions. That said, I still have several concerns regarding the following aspects:
>
> **Evaluation metrics** (W2): The authors propose three retrieval metrics: path recall, GPT-4 triplet recall, and answer recall. However, there seems to be a mismatch in how these metrics are applied to the baseline methods (e.g., RoG in Figure 3). For example, RoG and GNN-RAG achieve good answer recall, but their GPT-4 triplet recall is worse than GraphSAGE/cosine similarity. These inconsistencies may suggest that certain metrics favor specific methods and I think the authors may need to evaluation retrieval precision (i.e., how many retrieved triplets are irrelevant).
>
> Moreover, based on my review of Tables 1, 2, and Figure 3, it appears that Tables 1 and 2 report top-100 performance for SubgraphRAG, while the competing baselines (GNN-RAG, RoG) are evaluated at top-50. This discrepancy makes SubgraphRAG's performance seem inflated. If this observation is correct, I recommend aligning the evaluation, e.g., by reporting top-10 performance across all methods for a fairer comparison.
>
> Regarding Table 1, I also recommend clarifying how time overhead is reported. The authors mention precomputing node/relation embeddings (Lines 361-364), which could result in lower time overhead of SubgraphRAG, but with higher memory overhead. Moreover, the downstream LLM time response is not reported, and SubgraphRAG seems to use a larger input context of top-100. I suggest discussing the trade-offs between time and memory usage for SubgraphRAG, and how SubgraphRAG's retrieved context impacts LLM response times.
>
> Additionally, for the new downstream results, I would appreciate clarification on whether SubgraphRAG uses top-100 triplets and what the average/median input size for RoG and GNN-RAG is.
>
> **Scientific soundness** (W1): It remains unclear why the proposed triplet scoring method outperforms more advanced methods like GNNs (the explanation in Line 215 is vague). For example, as I understand it, GNNs are capable of traversing deeper into the graph to retrieve multi-hop answers, even when considering only the top-1 retrieved nodes. In contrast, SubgraphRAG is limited to retrieving only 1-hop triplets at the top position, which may restrict its performance on multi-hop questions. I would recommend that the authors clarify in which specific scenarios SubgraphRAG is more effective or advantageous compared to competing methods, such as GNNs.
>
> Once again, I appreciate the authors' thorough work. However, I am unable to increase my score at this time due to the concerns outlined above. I recommend that the authors refine the evaluation to ensure fairness and identify specific use cases where SubgraphRAG excels over more advanced baselines.

---

> ### Author Response · Authors · 2024-11-24
>
> Thank you for reading our responses and providing detailed feedback!
>
> **Q1**: It remains unclear why the proposed triple scoring method outperforms more advanced methods like GNNs (the explanation in Line 215 is vague). For example, as I understand it, GNNs are capable of traversing deeper into the graph to retrieve multi-hop answers, even when considering only the top-1 retrieved nodes. In contrast, SubgraphRAG is limited to retrieving only 1-hop triples at the top position, which may restrict its performance on multi-hop questions. I would recommend that the authors clarify in which specific scenarios SubgraphRAG is more effective or advantageous compared to competing methods, such as GNNs.
>
> **Response**: We appreciate this important question comparing SubgraphRAG's multi-hop capabilities to GNNs. SubgraphRAG’s effectiveness in multi-hop retrieval is fundamentally enabled by our **Directional Distance Encoding (DDE)** mechanism detailed in **Section 3.1**.  Retrieval for multi-hop reasoning fundamentally requires capturing both **distance relative to variable-sized topic entities** and semantic information. While GNNs are effective for various scenarios, they face theoretical limitations in capturing distance information, as shown in seminal GNN theory studies [1, 2]. Distance encoding has emerged as an effective solution to mitigate this issue by labeling nodes with binary features and performing multiple rounds of training-free message passing. Our DDE mechanism extends this idea specifically for KGQA by labeling topic entities, enabling efficient approximate computation of **relative distances both from and towards variable-sized topic entities**. When combined with semantic text embeddings, this approach proves particularly effective for triple scoring. **Our ablation studies with respect to retrieval size (# triples) and question hop in Table 2, Figure 3 and Figure 5 demonstrate the effectiveness of DDE.**
>
> We acknowledge that GNN research has a rich history spanning at least eight years since the introduction of GCNs [3], yet **their application to KG-based RAG remains relatively unexplored**. We position SubgraphRAG as a valuable GNN-inspired baseline to catalyze further research in both GNN-based and GNN-inspired approaches.
>
> [1] Li et al. Distance Encoding: Design Provably More Powerful Neural Networks for Graph Representation Learning
>
> [2] Zhang et al. Labeling Trick: A Theory of Using Graph Neural Networks for Multi-Node Representation Learning
>
> [3] Kipf & Welling. Semi-Supervised Classification with Graph Convolutional Networks
>
> **Q2**: Why do RoG and GNN-RAG fall short for GPT-4o triple recall despite their good performance for path and answer entity recall (Figures 3 and 5)? Notably, their performances are worse than the GraphSAGE variant of SubgraphRAG and cosine similarity.
>
> **Response**: Thank you for this insightful question. The performance gap you’ve identified reveals a fundamental limitation in path-based retrieval approaches like RoG and GNN-RAG while highlighting the advantages of SubgraphRAG’s **flexible subgraph retrieval**. Specifically, RoG fine-tunes an LLM to extract shortest paths between topic entities and answer entities, while GNN-RAG trains GNNs to predict answer entities and extracts shortest paths from topic entities to predicted answer entities.
>
> The core challenge in KG-based RAG lies in retrieving **sufficient subgraph evidence** for LLM reasoning, which requires **going beyond retrieving candidate answer entities or specialized subgraphs like shortest paths**. However, ground truth subgraphs are rarely available in practice. As a workaround, existing approaches like RoG use shortest paths as weak supervision signals for training retrievers. Recognizing the limitations of shortest paths in representing complete evidence subgraphs, we leverage GPT-4o to label relevant subgraphs for more comprehensive retrieval evaluation. While this labeling approach may not be perfect, it is currently the most viable option.
>
> The observed performance discrepancies between shortest path/answer entity retrieval and GPT-4o triple retrieval for RoG and GNN-RAG empirically validate the inherent limitations of path-based approaches. In contrast, SubgraphRAG variants and the cosine similarity baseline perform general triple retrieval and are not restricted to paths. **For fair comparison, we train SubgraphRAG variants with shortest path triples like RoG, but our retriever mechanism enables them to generalize beyond the specialized paths, resulting in superior performance in retrieving GPT-4o triples.**

---

> ### Author Response · Authors · 2024-11-24
>
> **Q3**: Figure 3 appears to be new. Based on my review of Tables 1, 2, and Figure 3, it appears that Tables 1 and 2 report top-100 performance for SubgraphRAG, while the competing baselines (GNN-RAG, RoG) are evaluated at top-50. This discrepancy makes SubgraphRAG's performance seem inflated. If this observation is correct, I recommend aligning the evaluation, e.g., by reporting top-10 performance across all methods for a fairer comparison.
>
> **Response**: We appreciate the reviewer's attention to evaluation details. However, the different retrieval sizes do not inflate SubgraphRAG's performance - they reflect **the optimal settings determined in the original papers**. The baseline results for RoG and GNN-RAG are reported using their officially recommended configurations. These recommended settings are often chosen based on optimal downstream evaluation performance. See for example Figure 3 in the RoG paper. Similarly, we determined top-100 as optimal for SubgraphRAG through validation with Llama 3.1-8B, the smallest language model in our evaluation suite.
>
> **Figures 3 and 5 originally only study the retrieval performance (recall @ K) of various SubgraphRAG variants across various retrieval sizes. We've updated it to include the baseline results.** The results show that SubgraphRAG **consistently outperforms the baseline retrievers when evaluated under identical retrieval size**. Artificially constraining well-performing methods to use suboptimal settings would not provide meaningful insights about their actual effectiveness for downstream LLM reasoning tasks.
>
> **Q4**: Regarding Table 1, I also recommend clarifying how time overhead is reported. The authors mention precomputing node/relation embeddings (Lines 361-364), which could result in lower time overhead of SubgraphRAG, but with higher memory overhead. Moreover, the downstream LLM time response is not reported, and SubgraphRAG seems to use a larger input context of top-100. I suggest discussing the trade-offs between time and memory usage for SubgraphRAG, and how SubgraphRAG's retrieved context impacts LLM response times.
>
> **Response**: Thank you for carefully checking the evaluation settings. We agree that time and memory trade-offs are critical considerations for deploying KG-based RAG systems in real-world scenarios. However, we would like to clarify that **embedding pre-computation is not unique to SubgraphRAG** – it is a common requirement shared across most baselines in our evaluation, including cosine similarity, G-Retriever, and GNN-RAG. To ensure fair comparison, we consistently excluded embedding pre-computation time across all these approaches. We’ve updated Lines 357-361 to provide a more precise description of wall clock time measurements and include a discussion of the time-memory trade-offs.
>
> Regarding the impact of context size on LLM response time, empirical studies demonstrate that the overall question-answering latency is hierarchically influenced by: (1) the number of LLM calls (primary factor), (2) output token count (secondary factor), and (3) input token count (tertiary factor). For LLMs like GPT-4 and Claude-3.5, an additional input token typically adds about $10^{-2}$ the latency of an extra output token and $10^{-4}$ that of an additional LLM call [1]. While SubgraphRAG may have varying input token counts, it maintains efficiency by requiring only a single LLM call per question, in contrast to baselines that necessitate multiple calls like RoG and ToG.
>
> [1] Vivek. Understanding input/output tokens vs latency tradeoff.
>
> **Q5**: For the new downstream results, I would appreciate clarification on whether SubgraphRAG uses top-100 triplets and what the average/median input size for RoG and GNN-RAG is.
>
> **Response**: The average input size for RoG and GNN-RAG are visualized in Figures 3 and 5 while SubgraphRAG employs top-100. To thoroughly investigate retrieval size effects, we conducted experiments by varying the beam search count for RoG, which directly influences the number of retrieved paths and triples. Our controlled comparisons between RoG's retriever and our approach, using **identical LLM reasoners (Llama 3.1-8B and GPT-4-mini) and equivalent numbers of retrieved triples**, demonstrate that **SubgraphRAG's superior retrieval mechanism translates directly into enhanced downstream reasoning performance, as evidenced in Figure 6**.

---

> > ### Author Response · Authors · 2024-11-25
> >
> > Dear Reviewer y2dZ,
> >
> > As the discussion period ends soon, we would like to check whether our responses answer your questions. To address your follow-up questions, we have clarified about the source of SubgraphRAG's multi-hop retrieval capability, the rationale behind the observed retrieval performance discrepancy with respect to different targets, the effects of retrieval size in our evaluations, and the retrieval efficiency measurement. Thank you again for your comments and suggestions to improve our paper, and we look forward to your reply.
> >
> > Best, Authors

---

> > > ### Comment · Reviewer_y2dZ · 2024-11-27
> > >
> > > Thank you for your response and for continuing this discussion. Below are my further comments and concerns:
> > >
> > > Q1) I don't observe significant performance differences between MLP, MLP+entity, and Subgraph in the entity recall metrics, which raises the concern that SubgraphRAG may not effectively leverage the graph structure. Furthermore, the use of GNNs in KGQA has been well-explored in prior work (e.g., [1]), which can be extended to augmenting GNN predictions with a LLM on top. \
> > > [1] Sun, Haitian, et al. "Open domain question answering using early fusion of knowledge bases and text." arXiv preprint arXiv:1809.00782 (2018).
> > >
> > > Q2) It remains unclear why Subgraph scores outperform RoG in GPT-4o-triplets, especially given that both methods use the same supervisory signals and they score similar performance in entity recall. I believe that certain evaluation metrics may inherently favor particular approaches.
> > >
> > > Q3) The term "optimal" setting is ambiguous. The definition of "optimality" can vary depending on the downstream LLM, which SubgraphRAG does not consider in the methodology. For instance, a smaller model might struggle with longer inputs (e.g., top-100). I recommend that the authors provide a more detailed evaluation across different input lengths (e.g., top-1, top-5, top-10) to clarify this.
> > >
> > > Q4) The authors' response does not provide numbers in practice regarding the latency cost of SubgraphRAG, especially given that it requires a longer context as input compared to more lightweight approaches like G-Retriever or GNN-RAG. The use of the term "efficient" in the title may not fully reflect practical latency overhead.
> > >
> > > Q5) Do the authors mean that RoG and GNN-RAG use top-50 as input, based on the figures provided? I would appreciate confirmation on this. In reviewing the original papers on RoG and GNN-RAG, I observed that they typically use fewer triplets (e.g., Table 5 in RoG and Figures 4-5 in GNN-RAG).
> > >
> > > Overall, I stand by my initial evaluation, as I believe my concerns are well-supported.

---

### Official Review · Reviewer_Yzun · 2024-10-31

**Soundness:** 3
**Presentation:** 3
**Contribution:** 2
**Rating:** 6
**Confidence:** 4

**Summary:**

This paper proposes SubgraphRAG, extending the KG-based RAG framework. To be specific, this method integrates a lightweight multilayer perceptron (MLP) with a parallel triple-scoring mechanism for efficient subgraph retrieval while enhancing retrieval accuracy by encoding directional structural distances. In addition, SubgraphRAG strikes a balance between model complexity and reasoning power as the size of retrieved subgraphs can be flexibly adjusted to match the query’s need and downstream LLM’s reasoning power. Extensive experiments demonstrate the effectiveness of the proposed methods.

**Strengths:**

1. The idea of employing lightweight models for retrieval and using large language models for reasoning is interesting, sensible, and intuitive.
2. The proposed method is efficient in terms of training time, requiring only the training of an MLP.

**Weaknesses:**

1. **May Lack novelty.** The novelty behind the proposed method may be insufficient as embedding-based retrieval methods for triplets [1] resemble existing approaches in the literature. Embedding-based retrieval methods for triplets utilize the embeddings of questions and triplets to retrieve the relevant triples. However, the proposed approach merely concatenates $z_\tau$ with the previous embeddings, which may lack sufficient innovation.
2. **Baseline methods.** As the proposed methods follow the retrieved-based paradigms, they would be better compared to more retrieved-based methods, such as UniKGQA [2], Subgraph Retrieval (SR) [3], and GNN-RAG [4], which are relevant to the proposed method and derive new SOTA for this task.
3. **Experimental Results.** As shown in Table 1 for the evaluation for retrieval recall, it would be better to present the number of extracted triplets in a new column for each baseline to ensure fair comparison. The same for Table 3 to demonstrate the QA performance.
4. In line 226, the paper claims that "GNNs are known to have limited representation power," which is why GNNs were not chosen for the retriever. However, as shown in Table 5, the paper presents different variants of SubgraphRAG by incorporating various retrieval methods. Including a GNN-based retrieval method for comparison would strengthen the evaluation, such as RevRea [5] as a GNN-based retriever.


[1] Li, Shiyang, et al. "Graph Reasoning for Question Answering with Triplet Retrieval." Findings of the Association for Computational Linguistics: ACL 2023. 2023.

[2] Jiang, Jinhao, et al. "UniKGQA: Unified Retrieval and Reasoning for Solving Multi-hop Question Answering Over Knowledge Graph." The Eleventh International Conference on Learning Representations.

[3] Zhang, Jing, et al. "Subgraph Retrieval Enhanced Model for Multi-hop Knowledge Base Question Answering." Proceedings of the 60th Annual Meeting of the Association for Computational Linguistics (Volume 1: Long Papers). 2022.

[4] Mavromatis, Costas, and George Karypis. "GNN-RAG: Graph Neural Retrieval for Large Language Model Reasoning." arXiv preprint arXiv:2405.20139 (2024).

[5] Mavromatis, Costas, and George Karypis. "ReaRev: Adaptive Reasoning for Question Answering over Knowledge Graphs." 2022 Findings of the Association for Computational Linguistics: EMNLP 2022. 2022.

**Questions:**

Please see in **Weaknesses** above.

---

> ### Author Response · Authors · 2024-11-22
>
> Thank you for carefully reading our manuscript and providing detailed constructive feedback. While we address your main questions below, we are actively working on the additional baseline comparisons you suggested.
>
> **Q1**: How is the evaluation of SubgraphRAG with respect to the baselines affected by the retrieval size (# triples)?
>
> **Response**: Thank you for the insightful question. While SubgraphRAG may utilize more triples in certain settings, our extensive experiments demonstrate that the performance improvements stem from substantial improvement in retrieval.
>
> Our analysis in Figures 3 and 5 presents retrieval recall values across varying numbers of retrieved triples. The results show that SubgraphRAG **consistently outperforms baseline retrievers when operating under identical retrieval budgets**. Importantly, SubgraphRAG demonstrates robust **scaling properties**: as we incorporate more top-ranked triples, the recall improvements remain substantial. This characteristic is particularly valuable for accommodating the varying long-context reasoning capabilities of different LLMs.
>
> For reasoning evaluation, we investigate the effects of adjusting beam search count for RoG [1], a competitive and reproducible baseline. The original RoG paper has already examined this aspect with its fine-tuned LLM, demonstrating **optimal reasoning performance at their chosen beam search count** (Figure 3, RoG paper). Our additional comparisons between the RoG retriever and our retriever, using identical LLM reasoners (Llama 3.1-8B and GPT-4o-mini) and the same numbers of retrived triples, reveal that SubgraphRAG’s retriever **maintains superior performance even when controlling the retrieval size** (Figure 6).
>
> Beyond the empirical studies, a fundamental design principle of this work is selecting a suitable number of triples that the LLM reasoner can process effectively, where suitability here is defined based on the reasoning capabilities of the specific LLMs. Notably, prior graph-based RAG methods, to the best of our knowledge, all overlook this crucial dimension.
>
> In our study, LLMs handle the slightly larger numbers of triples in Table 3 without difficulty. In fact, the larger numbers of triples reported in Table 3 are still below the maximum that certain LLMs can process. As shown in Figure 4, LLama-3.1 8B performs optimally with around 100–200 triples, while GPT-4o-mini continues to improve with up to 500 triples.
>
> This insight reflects the key contribution of our work: *emphasizing the retriver efficiency which enables the size-adjustable retrieval of a suitable (potentially larger) number of relevant triples via a lightweight model while leveraging the strong reasoning capabilities of LLMs to digest all those retrieved triples*. This principle is supported by the results presented in Table 3, Figure 4, and the retriever latency results in Table 1.
>
> **Q2**: How well does SubgraphRAG perform when compared to GNN-RAG?
>
> **Response**: For retrieval evaluation, our retriever consistently outperforms GNN-RAG (Table 1, Figure 3, and Figure 5). **Notably, while our retriever is not explicitly trained to predict answer entities as in GNN-RAG, it outperforms GNN-RAG in answer entity recall using equivalent retrieval budgets (Figure 3 and 5).** In terms of efficiency, our retriever is **an order of magnitude faster** than GNN-RAG (Table 1). For end-to-end KGQA evaluation, SubgraphRAG performs better or comparable with respect to GNN-RAG (Table 3).
>
> We have conducted additional ablation studies that isolate the retriever’s impact by controlling the LLM reasoner. Our experiments employ Llama 3.1-8B and more powerful GPT-4o-mini as fixed LLM reasoners while varying only the retrieval method. The results below demonstrate that SubgraphRAG consistently outperforms GNN-RAG across both LLM reasoners and datasets.
>
> **Llama 3.1-8B + WebQSP**
>
> | Retriever  | F1 | Hit |
> |--|----|---|
> | GNN-RAG | 62.34 | 78.81 |
> | SubgraphRAG | **70.57** | **86.61** |
>
> **Llama 3.1-8B + CWQ**
>
> | Retriever  | F1 | Hit |
> |--|----|---|
> | GNN-RAG | 46.40 | 54.35 |
> | SubgraphRAG | **47.16** | **56.98** |
>
> **GPT-4o-mini + WebQSP**
>
> | Retriever  | F1 | Hit |
> |--|----|---|
> | GNN-RAG | 73.27 | 85.93 |
> | SubgraphRAG | **77.45** | **90.11** |
>
> **GPT-4o-mini + CWQ**
>
> | Retriever  | F1 | Hit |
> |--|----|---|
> | GNN-RAG | 52.91 | 59.64 |
> | SubgraphRAG | **54.13** | **62.02** |
>
> Overall, the empirical studies demonstrate the surprising effectiveness and efficiency of our approach when compared to dedicated GNN-based approaches. The performance gains may be attributed to fundamental limitations of GNNs, including limited representation power for structural information [2] and degradation of pre-trained text embeddings due to over-smoothing [3] and over-squashing [4].
>
> We’ve updated our manuscript to properly include all the above results and properly cited ReaRev, the GNN architecture employed by GNN-RAG.

---

> > ### Author Response · Authors · 2024-11-22
> >
> > **Q3**: What’s the novelty of the proposed approach, when compared to the existing approach that performs embedding-based triple retrieval [5]?
> >
> > **Response**: Our work advances beyond [5] in a fundamental way. We introduce the first **systematic investigation of size-adjustable subgraph retrieval in KG-based RAG**, an advancement particularly relevant given the emerging focus on long-context language models and their application for RAG [6, 7]. This differs substantially from [5]'s **traditional natural language understanding setting**.
> >
> > Second, we address a critical limitation in [5]'s approach: their retrieval mechanism operates **without awareness of structural dependencies** among triples, making it inadequate for questions requiring **multi-hop reasoning or involving multiple topic entities**. Our DDE method explicitly captures these structural dependencies while maintaining computational efficiency. The empirical results validate these contributions, with SubgraphRAG consistently outperforming structure-unaware baselines (Table 1, Figure 3, Figure 5) and demonstrating superior performance on complex multi-hop and multi-topic-entity questions (Table 2, 7).
> >
> > [1] Luo et al. Reasoning on Graphs: Faithful and Interpretable Large Language Model Reasoning.
> >
> > [2] Li et al. Distance Encoding: Design Provably More Powerful Neural Networks for Graph Representation Learning.
> >
> > [3] Oono & Suzuki. Graph Neural Networks Exponentially Lose Expressive Power for Node Classification.
> >
> > [4] Topping et al. Understanding over-squashing and bottlenecks on graphs via curvature.
> >
> > [5] Li et al. Graph Reasoning for Question Answering with Triplet Retrieval. Findings of the Association for Computational Linguistics: ACL 2023. 2023.
> >
> > [6] Leng et al. Long Context RAG Performance of Large Language Models. 2024.
> >
> > [7] Yue et al. Inference Scaling for Long-Context Retrieval Augmented Generation. 2024.

---

> ### Author Response · Authors · 2024-11-25
>
> **Q4**: How well does SubgraphRAG perform when compared to SR [1] and UniKGQA [2]?
>
> **Response**: We conducted a thorough comparison with SR + NSM w E2E, which was identified in [1] as one of the strongest performing variants across both datasets. We have updated our manuscript to reflect these new evaluation results.
>
> Regarding **reproducibility**, we successfully executed comparisons using the official codebase for the WebQSP dataset. However, we were not able to run the codebase for the CWQ dataset, an issue that has been reported and remained open in SR's official GitHub repo after more than two years (https://github.com/RUCKBReasoning/SubgraphRetrievalKBQA/issues/6). As a result, for CWQ, we directly referenced the KGQA performance reported in the original paper.
>
> For **retrieval evaluation**, SubgraphRAG achieves **superior performance across both effectiveness and efficiency, as evidenced in Tables 1 and 2**. This performance advantage persists when **controlling for retrieval size, as illustrated in Figure 5**. Furthermore, in terms of **end-to-end KGQA performance**, SubgraphRAG **consistently outperforms** SR + NSM w E2E across all evaluated metrics.
>
> For UniKGQA, we were not able to run the official codebase due to various issues. For example, the link to the preprocessed data and KG dump in the README file is simply empty: https://github.com/JBoRu/UniKGQA. Multiple unresolved GitHub issues further document these accessibility problems. Given these constraints, we have included UniKGQA's originally reported KGQA performance metrics in Table 3, which still demonstrate SubgraphRAG's superior performance.
>
> [1] Zhang, Jing, et al. "Subgraph Retrieval Enhanced Model for Multi-hop Knowledge Base Question Answering." Proceedings of the 60th Annual Meeting of the Association for Computational Linguistics (Volume 1: Long Papers). 2022.
>
> [2] Jiang, Jinhao et al. "UniKGQA: Unified Retrieval and Reasoning for Solving Multi-hop Question Answering Over Knowledge Graph." ICLR 2023.

---

> ### Author Response · Authors · 2024-11-25
>
> Dear Reviewer Yzun,
>
> As the discussion period ends soon, we would like to check whether our responses answer your questions. Following your suggestions, we have clarified the effects of retrieval size in our evaluations, incorporated three new baselines (GNN-RAG, SR, UniKGQA), and highlighted our contributions when compared to a previous work. Thank you again for your comments and suggestions to improve our paper, and we look forward to your reply.
>
> Best, Authors

---

> > ### Comment · Reviewer_Yzun · 2024-11-26
> >
> > I appreciate the author's response to my concerns.
> >
> > However, I remain curious about the generalizability of the proposed method, specifically whether the retriever needs to be retrained when applied to a new knowledge graph (KG). While the revised paper demonstrates generalizability from WebQSP to CWQ in Table 3, both datasets are based on Freebase. It would be valuable to extend this evaluation to other KG datasets, such as CommonsenseQA, which is based on ConceptNet.
> >
> > Therefore, I will maintain my current score and look forward to the author's responses to these questions.

---

> > > ### Author Response · Authors · 2024-11-26
> > >
> > > Thank you for the insightful comments regarding generalizability. Generalizability is actually what we are now investigating. Our preliminary results show **promising cross-dataset generalization between WebQSP and CWQ, achieving better performance than more complex SOTA like GNN-RAG (Tables 1, 6, and 9)**. We attribute this improved generalization to two **key design choices**: (1) the use of pre-trained text embeddings that capture transferable semantic information, and (2) simple MLP architecture that reduces overfitting to dataset-specific patterns. We acknowledge that the current generalization performance, while encouraging, still has room for improvement. We are actively investigating approaches to further enhance cross-dataset and cross-KG transfer. It's worth noting that **achieving strong generalization across different KGs remains an open challenge in the field - to the best of our knowledge, even state-of-the-art training-based methods have not demonstrated generalization performance comparable to ours within the Freebase KG family**. We appreciate your suggestion about evaluation on CommonsenseQA/ConceptNet, which would indeed provide valuable insights into cross-KG generalization. We plan to explore this direction in future work.

---

> > > > ### Comment · Reviewer_Yzun · 2024-11-27
> > > >
> > > > Thank you for your response. However, I am still curious about the generalization of the proposed methods.
> > > >
> > > > Therefore, I will maintain my current score.

---

### Official Review · Reviewer_Jjux · 2024-11-03

**Soundness:** 3
**Presentation:** 2
**Contribution:** 3
**Rating:** 5
**Confidence:** 5

**Summary:**

This manuscript introduces a framework called SubgraphRAG, which is a KG-based RAG system designed to enhance the reasoning capabilities of LLMs by integrating structured information from knowledge graphs.

-  The framework employs a lightweight multilayer perceptron (MLP) combined with a parallel triple scoring mechanism, proposing to adopt a retriever that allows for subgraph distribution factorization. This means that each triple can be optimized independently, which improves the efficiency of subgraph retrieval.
-  By encoding directional structural distances as structural features using DDE, the accuracy of retrieval is enhanced.

**Strengths:**

- Efficiency and Scalability: As the authors present in Table 1, SubgraphRAG demonstrates high efficiency and scalability in the retrieval process. Additionally, the flexible form of retrieved subgraphs, with adjustable sizes—subgraphs formed by top-triples can accommodate various LLMs with diverse reasoning capabilities.
- By combining DDE and MLP, SubgraphRAG surpasses more complex models, such as RoG and G-Retriever, in terms of covering key triples and entities.

**Weaknesses:**

- I affirm: The optimization objective defined in the article states that the SubgraphRAG retriever distribution Qθ can be factorized into distributions over triples, which means that the retrieved subgraphs do not necessarily have to follow a fixed type (such as trees or paths). This design allows for efficient training, efficient subgraph retrieval, flexible subgraph types, and adjustable subgraph sizes, rather than relying solely on path retrieval. This differs from the methods widely discussed in the current research community for finding retrieval triples and performing graph RAG, such as RoG and G-Retriever, which the article compares.

  - However，The main experiments require more graph retrieval methods, especially path retrieval, for comparison. For example:

  > Haoran Luo,et al., ChatKBQA: A Generate-then-Retrieve Framework for Knowledge Base Question Answering with Fine-tuned Large Language Models. ACL (Findings) 2024

  > Guanming Xiong, et al., Interactive-KBQA: Multi-Turn Interactions for Knowledge Base Question Answering with Large Language Models. ACL (1) 2024


  - I am also curious as to why the ToG method is not reproducible, as it seems to be a strong contender in the Hit results. The calculation of ToG's hit@1 is actually based on ToG only finding the first match in the answer text, which is considered as the answer by the large model to calculate Hits@1. However, I do not believe this is the reason for the irreproducibility. Interactive-KBQA has reproduced ToG's results.


[Subtle weaknesses]

- It was not clearly stated whether the experimental results of comparison methods such as RoG and G-Retriever were reproduced a second time or referenced from the original text. This is because the referenced StructGPT missed many tests, while the results of baseline methods like RoG should have been reproduced by the authors themselves.

[Subtle weaknesses]

- Typos and Formatting Issues
  - Naming  Eq.3 and q)).
  - line 052 ``
  - line 161 .
  - line 986 .
  - line 966 ，968，970 .
- There are some grammar errors, not listed one by one

**Questions:**

- Some Ambiguous Expressions Lack Further Mathematical Definition or Explanation
  - What is ‘scalable’ and how to prove whether the method is ‘scalable’ or not?
  - What constitutes suitable prompting?

- The authors noted that baselines report Hit@1 but compute Hit, assessing if any correct answer is in the LLM's response. There's also confusion over metrics like extract match and Hit@1 in the research community. Clarification of these metrics in the appendix is needed, and defining another "Hit metric" is not advised.

---

> ### Author Response · Authors · 2024-11-22
>
> Thank you for carefully reading our manuscript and providing detailed and valuable feedback. Below we respond to your questions and concerns.
>
> **Q1**: How well does the SubgraphRAG retriever perform when compared to retrievers in ChatKBQA [1] and Interactive-KBQA [2]?
>
> **Response**: Thank you for suggesting these additional retrieval baselines to help us strengthen our work. We have already updated our manuscript to cite these works. We fully agree that experimental comparisons with these approaches would provide valuable insights. We are actively conducting these additional experiments and commit to updating our results with these comparisons before the end of the rebuttal period.
>
> **Q2**: Are the experiment results of the baselines reproduced a second time or referenced from the original text? In particular, it seems that the referenced StructGPT missed many tests, while the results of baseline methods like RoG should have been reproduced by the authors themselves.
>
> **Response**: We appreciate the reviewer's attention to our experiment settings. Our baseline results were obtained through a combination of reproduction and citation of original results. For retrieval evaluations, we provide comprehensive details in Appendix B.1. For reasoning evaluations, we include discussions in the “Experiment Settings” in Sec. 4.2. For baseline methods such as RoG, we managed to fully reproduce their results using the official pre-trained models and codebase.
>
> Regarding StructGPT, the selective inclusion was based on several practical considerations. First, WebQSP was the only dataset in our study for which StructGPT reported results. Second, the authors provided processed files specifically for WebQSP, facilitating accurate comparison. Third, reproducing StructGPT's results for other datasets proved challenging, as evidenced by numerous unresolved reproduction issues in their official GitHub repository. Finally, given that StructGPT no longer represents the state-of-the-art performance in this domain, we strategically focus our reproduction efforts on more recent and competitive baseline methods.
>
> **Q3**: Can you clarify why the results of ToG are not reproducible? The calculation of ToG's Hits@1 is actually based on ToG only finding the first match in the answer text. However, this is not the reason for irreproducibility. Interactive-KBQA has reproduced results.
>
> Response: We attempted to run the official ToG codebase by following their README instructions, but we were unable to reproduce the results. While we have full confidence in the authors' scientific integrity, additional documentation and guidance would enhance the reproducibility of their work.
>
> Regarding the ToG results cited in the Interactive-KBQA paper [2], actually, their reproduced ToG results also differ substantially from those reported in the ToG paper. Specifically, ToG claims to be better than StructGPT by 10%, while the reproduced results from Interactive-KBQA show that ToG is worse than StructGPT by 8%. Therefore, the results of ToG from Interactive-KBQA essentially align with our observation that ToG’s reported results may not be easily reproducible given its released codebase.
>
> Thank you for pointing out that our original footnote caused confusion between metric choice and irreproducibility. We have updated the footnote to enhance clarity and will appreciate your guidance for further refinement.
>
> **Q4**: There are some ambiguous expressions that lack sufficient mathematical definitions or explanations. 1) What is “scalable” and how to prove whether the method is “scalable” or not? 2) What constitutes suitable prompting?
>
> **Response**: Thank you for these points of clarification. Regarding scalability, we consider both algorithmic time complexity and empirical wall clock time efficiency. Our method exhibits linear complexity in relation to the graph size/number of candidate triples. Empirically, it achieves improvements in wall clock time by orders of magnitude compared to the baselines (Table 1). Regarding suitable prompting, we investigated various prompting strategies and selected the optimal one based on empirical performance, with detailed results presented in Figure 2 and Appendix D.
>
> **Q5**: There are some typos, formatting issues, and grammar errors.
>
> **Response**: Thank you for the great efforts in proofreading our manuscript. We have revised the manuscript to fix the issues you explicitly pointed out. We will make a more careful pass of the manuscript after the rebuttal period for possible other issues.
>
> [1] Luo et al. ChatKBQA: A Generate-then-Retrieve Framework for Knowledge Base Question Answering with Fine-tuned Large Language Models. ACL (Findings) 2024.
>
> [2] Xiong et al. Interactive-KBQA: Multi-Turn Interactions for Knowledge Base Question Answering with Large Language Models. ACL 2024.

---

> > ### Author Response · Authors · 2024-11-25
> >
> > Dear Reviewer Jjux,
> >
> > As the discussion period ends soon, we would like to check whether our responses answer your questions. Thank you again for your comments and suggestions to improve our paper, and we look forward to your reply.
> >
> > Best, Authors

---

> > > ### Comment · Reviewer_Jjux · 2024-11-26
> > > **Maintain my score**
> > >
> > > Given the breadth of the experiments, I awarded this manuscript 5 points; however, the methodology did not fully captivate me. By combining DDE and MLP, SubgraphRAG outperforms more complex models. However, compared to approaches like RoG and ToG, which leverage knowledge graphs for explainable reasoning, this paper appears to sacrifice that aspect. This is an open issue, as achieving explainability often introduces additional reasoning costs for LLMs.
> > >
> > > Additionally, concerning my observations, it seems the KBQA research community would benefit from more standardized comparisons, especially with regard to evaluation metrics and LLM backbones. This, however, is not a flaw in the paper itself, and the authors have clarified some of the challenges related to reproducing current baselines in their rebuttal.
> > >
> > > In light of these points, I maintain my score.

---

> ### Author Response · Authors · 2024-11-26
>
> Thank you for reading our previous responses and recognizing **the breadth of our experiments**! We are glad that we’ve **addressed your previous questions** (e.g., the non-reproducibility of the ToG codebase) and have the opportunity to address your new questions.
>
> First, we have to say that whether a methodology is considered 'captivating' can be subjective. We find greater appeal in methods that are simple yet effective, rather than overly complex. That said, we prefer to prioritize **objective evaluation results as a measure of a method's merit**.
>
> **Q6**: With lightweight subgraph retrieval, does SubgraphRAG sacrifice explainable reasoning when compared to approaches like RoG and ToG. In particular, achieving explainability often introduces additional reasoning costs for LLMs.
>
> **Response**: Thank you for raising this important consideration. Actually, our manuscript has already demonstrated SubgraphRAG’s explainability and reasoning capabilities in Section 4.2. **Following the evaluation in the RoG paper, we have provided nine detailed explainable reasoning examples from WebQSP and CWQ in Appendix E**, which demonstrates the effectiveness of SubgraphRAG in providing explanations along with answer predictions. We understand that these explainability studies might have been overlooked given our broad experiment results, and we appreciate the opportunity to highlight them further.
>
> In contrast, RoG’s LLM fine-tuning approach potentially compromises the inherent explainable reasoning capabilities of LLMs, as explained in page 18, A.4 of the RoG paper (https://arxiv.org/abs/2310.01061). To compensate for this lost capability, RoG employs ChatGPT to label explanations for 2000 samples as supplementary fine-tuning data.
>
> Once again, our above comparisons are based on objective evidence rather than subjective impressions 'This paper appears to sacrifice that aspect'

---

> > ### Comment · Reviewer_Jjux · 2024-11-26
> > **Reply to the rebuttal.**
> >
> > - I recognize the `interpretability’ of methods like RoG and ToG, which ground and detect relation paths and reasoning paths to enable reasoning processes based on KG nodes.
> >
> > - Relying on the retrieval of a larger number of triplets (e.g., top-100) and feeding them into large models, diminishes the interpretability tied to reasoning paths.
> >
> > - Additionally, I noticed that the paper frequently claims to yield “significant” improvements (e.g., lines 114, 246, 518, 523, 975, 1133). I recommend conducting statistical significance tests (e.g., p-tests) to substantiate these claims relative to the baseline methods.

---

### Official Review · Reviewer_A8Bg · 2024-11-04

**Soundness:** 3
**Presentation:** 3
**Contribution:** 3
**Rating:** 6
**Confidence:** 4

**Summary:**

This paper introduces the SubgraphRAG framework for the first time, a knowledge-graph (KG)-based generation method specifically designed to optimize the efficiency and accuracy of large language models (LLMs) in retrieval-augmented generation (RAG) tasks. By incorporating a lightweight multilayer perceptron (MLP) and a parallel triple-scoring mechanism, SubgraphRAG efficiently retrieves subgraphs relevant to the query and improves retrieval precision through Directional Distance Encoding (DDE). This method effectively balances retrieval accuracy and computational complexity, achieving adaptive subgraph retrieval tailored to different LLM reasoning capabilities for the first time.

**Strengths:**

**Quality**: The experimental design is comprehensive, covering two major multi-hop datasets in KG-based question answering tasks (WebQSP and CWQ) and providing detailed comparisons with multiple baseline methods. Results show that SubgraphRAG outperforms existing baselines across several metrics, demonstrating higher retrieval efficiency and answer accuracy, particularly in multi-hop reasoning and complex structure retrieval tasks. Additionally, ablation studies validate the independent contributions and effectiveness of each component, such as DDE and MLP.

**Clarity**: The paper is well-structured, with clear explanations progressing from the background of KG-augmented generation tasks to the design of each module within SubgraphRAG. Not only does the paper provide a flowchart illustrating the SubgraphRAG framework, but it also explains each step’s design principles and implementation details in depth. Moreover, the experimental section offers a detailed analysis of different design variations, which helps readers understand the role of each component.

**Significance**: SubgraphRAG provides an innovative and efficient retrieval-reasoning method for KG-augmented generation tasks. Compared to existing methods, SubgraphRAG achieves strong generalizability and extensibility through flexible subgraph retrieval strategies and LLM reasoning without fine-tuning. It offers more robust support for knowledge-driven generation tasks, contributing to the broader and deeper application of KGs in practical generation tasks.

**Weaknesses:**

1. **Insufficient Exploration of LLMs’ Potential for Retrieval Support**: SubgraphRAG primarily relies on an MLP and Directional Distance Encoding (DDE) for subgraph retrieval, yet LLMs inherently excel in handling complex semantics and relational structures. Directly assigning the retrieval process to LLMs could improve flexibility in multi-hop reasoning scenarios and adaptivity to complex KG structures, potentially enhancing the framework’s ability to handle diverse types of queries.

2. **The Limitation of Topic Entity Bias on Generalizability**: The paper assumes that all queries can be guided by topic entity-induced inductive biases for retrieval. However, this topic-driven approach may restrict information coverage or introduce noise in cases involving ambiguous or polysemous queries. Relying on topic entities might lead to suboptimal performance for non-topic-focused queries.

3. **Limitations of Structured Triples**: SubgraphRAG presents triples to LLMs in structured form, rather than transforming them into natural language descriptions, which misses an opportunity to leverage LLMs’ strengths in natural language understanding and reasoning. Converting triples to natural language may allow LLMs to more fully exploit the triple information and achieve more accurate reasoning.

4. **Lack of a Complete Retrieval Process Example**: Although the paper details the algorithmic flow of SubgraphRAG, it lacks a full example from query to retrieval and reasoning to answer generation. Providing a concrete example, such as with a typical query from WebQSP or CWQ, would enhance readers’ understanding of the procedural details.

5. **Limited Experimental Dataset Scope**: Experiments are conducted solely on WebQSP and CWQ datasets, leaving out other mainstream multi-hop KGQA benchmarks, such as HotpotQA. A broader range of dataset tests could further verify the method’s generalizability and robustness across various KGQA tasks.

**Questions:**

1. Could LLM-driven retrieval better identify information required for multi-hop reasoning?

2. Does topic entity bias lead to misleading results in ambiguous queries, and is there a more generalized retrieval strategy available?

3. Should triples be directly converted to natural language for LLMs to better understand and utilize them?

4. Could a complete example using a typical query illustrate the entire retrieval-reasoning workflow of SubgraphRAG?

5. How does SubgraphRAG perform on other mainstream KGQA datasets, and is it equally effective?

---

> ### Author Response · Authors · 2024-11-21
>
> Thank you for your insightful and positive feedback! Below we respond to your questions and concerns.
>
> **Q1**: Although the paper details the algorithmic flow of SubgraphRAG, it lacks a full real example from query to retrieval and reasoning to answer generation. Could you provide a complete example for a typical concrete query from WebQSP or CWQ?
>
> **Response**: Thank you for helping us further improve the presentation. We’ve included examples for multiple real samples from WebQSP and CWQ in Appendix E, covering the corresponding question, retrieved triples, LLM responses, and ground truth answers.
>
> **Q2**: SubgraphRAG employs an MLP with pre-trained text embeddings and Directional Distance Encoding (DDE) for subgraph retrieval. As LLMs may inherently excel in processing complex semantic and relational information, could LLM-driven retrieval better identify information required for multi-hop reasoning?
>
> **Response**: The direct use of LLMs for retrieval, while theoretically promising, faces significant practical limitations. As demonstrated in Table 1, a single call of even a small 7B LLM (RoG [1]) is **two orders of magnitude more costly** than our retriever in terms of wall clock time efficiency. This efficiency gap widens further when considering the limited context window sizes of LLMs, which necessitate multiple calls – for instance, ToG [2] requires three separate calls for its iterative graph exploration approach.
>
> The challenges extend beyond mere computational efficiency. Even in cases where the context window size is sufficient, LLMs can still **struggle to reason over a long context** [3, 4]. Our ablation studies (Figure 4) provide empirical evidence that including more than 100 triples actually deteriorates the reasoning performance of Llama 3.1-8B.
>
> We acknowledge that complex reasoning scenarios might benefit from leveraging LLM capabilities through **multiple rounds of retrieval and reasoning**. This could be achieved by decomposing queries into smaller subqueries and applying our retriever to each component separately, thereby maintaining the efficiency advantages of our approach while potentially accessing the sophisticated reasoning capabilities of LLMs in a more controlled manner. This is beyond the scope of this work and we are actually currently working on this.
>
> **Q3**: The mechanism of DDE heavily relies on the assumption of high-quality topic entities. In practice, the extracted topic entities can be noisy or simply missing for ambiguous and polysemous queries. Under such circumstances, will the topic entity bias lead to misleading retrieval results? Is there a more generalized retrieval strategy available?
>
> **Response**: We greatly appreciate your insightful observation regarding DDE’s reliance on topic entities. While our current implementation demonstrates strong performance with high-quality topic entities, we acknowledge the challenges posed by ambiguous and polysemous queries. To address these challenges, we propose extending our approach through two complementary strategies.
>
> First, we can employ LLMs as a preprocessing step for **query refinement and disambiguation prior to topic entity extraction**. For instance, given an ambiguous query like "Tell me about Python's applications", the LLM can generate distinct interpretations: "What are the applications of Python, the programming language?" and "What are the characteristics of pythons, the snake species?". This disambiguation step allows our system to extract precise topic entities (Python [programming language] and Python [snake genus]) and provide accurate retrieval results for both interpretations.
>
> Second, when direct topic entity extraction proves challenging, we propose leveraging LLMs to **generate multiple candidate topic entities with associated confidence scores**. The retrieval results can then be aggregated across these candidates, weighted by their confidence scores, to provide more robust results. This approach maintains the benefits of entity-guided retrieval while being more resilient to noise in entity extraction.
>
> It's worth noting that this challenge is **not unique to DDE** – to the best of our knowledge most existing KGQA and KG-based RAG approaches, including our baselines Retrieve-Rewrite-Answer [5], RoG [6], and G-Retriever [7], similarly rely on extracted topic entities. Overall, the proposed extensions advance toward more robust entity-guided retrieval while preserving DDE's core advantages.

---

> ### Author Response · Authors · 2024-11-21
>
> **Q4**: SubgraphRAG presents triples to LLMs in structured form, i.e., (h, r, t), rather than transforming them into natural language descriptions, which misses an opportunity to leverage LLMs’ strengths in natural language understanding and reasoning. Converting triples to natural language may allow LLMs to more comprehensively exploit the triple information and achieve more accurate reasoning. Should triples be directly converted to natural language for LLMs to better understand and utilize them?
>
> **Response**: Thank you for the highly insightful suggestions. We have conducted empirical investigations comparing both approaches - structured triples versus natural language descriptions. Our experiments revealed **no clear performance differences** between the two representations. We suspect that structured triple representations post less severe challenges for long-context reasoning because they are **more compact in terms of the number of occupied tokens**. The compact nature of structured triples allows more information to be processed within the same context window compared to their natural language counterparts. Moreover, converting triples to natural language descriptions introduces additional computational overhead during LLM inference. These considerations led us to adopt structured representations in our approach, aligning with several contemporary graph-based RAG approaches [5, 6, 7], despite that some prior work has explored converting triples into natural language descriptions [8].
>
> **Q5**: Experiments are conducted solely on WebQSP and CWQ datasets. How does SubgraphRAG perform on other datasets like HotpotQA?
>
> **Response**: Thank you for this important question. We selected WebQSP and CWQ as our primary evaluation datasets because they represent the current standard benchmarks in KG-based RAG research, allowing direct comparisons with existing methods. While we acknowledge that evaluating on additional datasets would strengthen our findings, our approach is fundamentally **dataset-agnostic and can be applied to various knowledge graph question answering scenarios**. For instance, CommonsenseQA [9] and OpenBookQA [10], which utilize the ConceptNet knowledge graph [11] and also allow extracting topic entities from questions, present natural extension opportunities for SubgraphRAG. Regarding HotpotQA [12], though it is not inherently a KGQA dataset, efforts like [13] have demonstrated successful transformation of text knowledge base (Wikipedia) into structured graph representations. Such graph-based transformations of Wikipedia articles make HotpotQA compatible with our approach, where paragraphs relevant to a question are essentially topic entities. We agree that expanding our evaluation to these datasets would be valuable for future work.
>
>
> [1] Luo et al. Reasoning on Graphs: Faithful and Interpretable Large Language Model Reasoning.
>
> [2] Sun et al. Think-on-Graph: Deep and Responsible Reasoning of Large Language Model on Knowledge Graph.
>
> [3] Leng et al. Long Context RAG Performance of Large Language Models.
>
> [4] Liu et al. Lost in the Middle: How Language Models Use Long Contexts.
>
> [5] Wu et al. Retrieve-Rewrite-Answer: A KG-to-Text Enhanced LLMs Framework for Knowledge Graph Question Answering.
>
> [6] Luo et al. Reasoning on Graphs: Faithful and Interpretable Large Language Model Reasoning.
>
> [7] He et al. G-Retriever: Retrieval-Augmented Generation for Textual Graph Understanding and Question Answering.
>
> [8] Li et al. Graph Reasoning for Question Answering with Triplet Retrieval.
>
> [9] Talmor et al. CommonsenseQA: A Question Answering Challenge Targeting Commonsense Knowledge.
>
> [10] Mihaylov et al. Can a Suit of Armor Conduct Electricity? A New Dataset for Open Book Question Answering.
>
> [11] Speer et al. ConceptNet 5.5: An Open Multilingual Graph of General Knowledge.
>
> [12] Yang et al. HotpotQA: A Dataset for Diverse, Explainable Multi-hop Question Answering.
>
> [13] Asai et al. Learning to Retrieve Reasoning Paths over Wikipedia Graph for Question Answering.

---

> > ### Author Response · Authors · 2024-11-25
> >
> > Dear Reviewer A8Bg,
> >
> > As the discussion period ends soon, we would like to check whether our responses answer your questions. Thank you again for your comments and suggestions to improve our paper, and we look forward to your reply.
> >
> > Best,
> > Authors

---

> > > ### Comment · Reviewer_A8Bg · 2024-11-26
> > >
> > > Thank the authors for addressing my concerns and questions. After reviewing your responses, I have decided to maintain my original score, as it already reflects the contribution of the work.

---

> > > > ### Author Response · Authors · 2024-11-26
> > > >
> > > > Thank you again for your great efforts in reviewing our manuscript!

---

### Comment · Area_Chair_e8YF · 2024-11-25
**Please reply to the authors' response.**

Dear reviewers,

The ICLR author discussion phase is ending soon. Could you please review the authors' responses and take the necessary actions? Feel free to ask additional questions during the discussion. If the authors address your concerns, kindly acknowledge their response and update your assessment as appropriate.


Best,
AC

---

### Meta-Review · Area_Chair_e8YF · 2024-12-17

**Metareview:**

The paper introduces SubgraphRAG, a framework designed to enhance retrieval-augmented generation by integrating a lightweight multilayer perceptron (MLP) with a parallel triple-scoring mechanism for efficient subgraph retrieval. The framework aims to balance retrieval accuracy and efficiency, improving response grounding and reducing hallucinations. The paper demonstrates its strengths, including efficiency and scalability in retrieval processes, flexibility in adjusting subgraph sizes to match different LLM reasoning capabilities, and strong performance in retrieval efficiency and answer accuracy, particularly in multi-hop reasoning tasks.

However, there are concerns about the method's generalizability across different knowledge graphs and datasets, and the evaluation metrics used. Reviewers suggested including more metrics like precision and F1 score for a fairer comparison and addressing issues with reproducing results of some baseline methods like ToG. Detailed evaluations across different input lengths and practical latency costs were also recommended. The authors have provided detailed responses to reviewers' concerns, including additional experiments and comparisons with new baselines. Still, they are encouraged to include more discussion or experiments on the retrieval size effects (e.g., reducing the number of retrieval triplets) in the final manuscript.

Overall, the reviewers' comments indicate that the pros outweigh the cons, leading to a recommendation for acceptance.

**Additional Comments On Reviewer Discussion:**

There was an extensive discussion on the paper. There are concerns about the method's generalizability across different knowledge graphs and datasets, and the evaluation metrics used. Reviewers suggested including more metrics like precision and F1 score for a fairer comparison and addressing issues with reproducing results of some baseline methods like ToG. Detailed evaluations across different input lengths and practical latency costs were also recommended. The authors have provided detailed responses to reviewers' concerns, including additional experiments and comparisons with new baselines.

---

### Decision · Program_Chairs · 2025-01-22

Accept (Poster)